

# Global mapping of potential natural vegetation: an assessment of machine learning algorithms for estimating land potential

Tomislav Hengl[1], Markus G. Walsh[2,3], Jonathan Sanderman[4], Ichsani Wheeler[1], Sandy P. Harrison[5] and Iain C. Prentice[6]

[1] Envirometrix Ltd, Wageningen, Netherlands
[2] The Earth Institute, Columbia University, New York, NY, USA
[3] Selian Agricultural Research Institute, Arusha, Tanzania
[4] Woods Hole Research Center, Falmouth, MA, USA
[5] School of Archeology, Geography and Environmental Science, University of Reading, Reading, UK
[6] Department of Life Sciences and Grantham Institute—Climate Change and the Environment, Imperial College London, London, UK

Corresponding author
Tomislav Hengl,
tom.hengl@envirometrix.net

## ABSTRACT

Potential natural vegetation (PNV) is the vegetation cover in equilibrium with climate, that would exist at a given location if not impacted by human activities. PNV is useful for raising public awareness about land degradation and for estimating land potential. This paper presents results of assessing machine learning algorithms—neural networks (nnet package), random forest (ranger), gradient boosting (gbm), K-nearest neighborhood (class) and Cubist—for operational mapping of PNV. Three case studies were considered: (1) global distribution of biomes based on the BIOME 6000 data set (8,057 modern pollen-based site reconstructions), (2) distribution of forest tree taxa in Europe based on detailed occurrence records (1,546,435 ground observations), and (3) global monthly fraction of absorbed photosynthetically active radiation (FAPAR) values (30,301 randomly-sampled points). A stack of 160 global maps representing biophysical conditions over land, including atmospheric, climatic, relief, and lithologic variables, were used as explanatory variables. The overall results indicate that random forest gives the overall best performance. The highest accuracy for predicting BIOME 6000 classes (20) was estimated to be between 33% (with spatial cross-validation) and 68% (simple random sub-setting), with the most important predictors being total annual precipitation, monthly temperatures, and bioclimatic layers. Predicting forest tree species (73) resulted in mapping accuracy of 25%, with the most important predictors being monthly cloud fraction, mean annual and monthly temperatures, and elevation. Regression models for FAPAR (monthly images) gave an R-square of 90% with the most important predictors being total annual precipitation, monthly cloud fraction, CHELSA bioclimatic layers, and month of the year, respectively. Further developments of PNV mapping could include using all GBIF records to map the global distribution of plant species at different taxonomic levels. This methodology could also be extended to dynamic modeling of PNV,

so that future climate scenarios can be incorporated. Global maps of biomes, FAPAR and tree species at one km spatial resolution are available for download via http://dx.doi.org/10.7910/DVN/QQHCIK.

## INTRODUCTION

Potential natural vegetation (PNV) is the *"vegetation cover in equilibrium with climate, that would exist at a given location non-impacted by human activities"* (*Levavasseur et al., 2012*; *Hemsing & Bryn, 2012*). It is a hypothetical vegetation state assuming natural (undisturbed) physical conditions, a reference status of vegetation assuming no degradation and/or no unusual ecological disturbances. PNV is especially useful for raising public awareness about land degradation (*Weisman, 2012*) and for estimating land potential (*Herrick et al., 2013*). For example, *Omernik (1987)* details PNV maps for USA; *Bohn, Zazanashvili & Nakhutsrishvili (2007)* provides maps for EU; *Carnahan (1989)* for Australia; *Marinova et al. (2018)* maps PNV for the Eastern Mediterranean–Black Sea–Caspian-Corridor; and maps of PNV for Latin America are available in *Marchant et al. (2009)*. Regarding specific tree species, *San-Miguel-Ayanz et al. (2016)* provide habitat suitability maps for the main forest tree species in Europe, based on environmental variables, especially bioclimatic variables such as average temperature of the coldest month, precipitation of the driest month and similar. *Potapov, Laestadius & Minnemeyer (2011)* generated a global map of potential forest cover at one km resolution (publicly available from http://globalforestwatch.org/map/). *Erb et al. (2017)* produced a global map of potential biomass stocks by reversing the current managed land use systems to natural vegetation. *Levavasseur et al. (2012)* and *Tian et al. (2016)* predict global PNV classes using environmental covariates such as climatic images and landform parameters. *Griscom et al. (2017)* recently produced a global reforestation map at one km resolution.

A common limitation of existing maps is their coarse spatial resolution (about 25 km) limiting the use of these maps for operational planning (*Marchant et al., 2009*; *Levavasseur et al., 2012*; *Tian et al., 2016*). In addition, comparisons of multiple overlapping sources of PNV maps shows that they rarely agree with one another since they do not share the same mapping criteria and, traditionally, emphasize regionally-specific botanical groupings rather than functional classifications. Limitations of maps based on field surveys of PNV (*Bohn, Zazanashvili & Nakhutsrishvili, 2007*) arise from assumptions about controls on vegetation distribution based on extrapolation from a limited number of field surveys.

Here, we provide an update of comparable global PNV maps produced by *Potapov, Laestadius & Minnemeyer (2011)*, *Levavasseur et al. (2012)*, *Tian et al. (2016)*, and
*Erb et al. (2017)*. We explore the possibility of increasing the mapping accuracy using up-to-date maps of climate, atmosphere dynamics, landform and lithology, and state-of-the-art machine learning methods. Our final aim is to produce PNV maps that are more detailed, richer in information, based on objective reproducible methods; and potentially more usable for global modeling and awareness raising projects. We focus on improving the spatial detail, thematic accuracy, and reproducibility of maps, at the cost of increasing the total computing load. We also consider automation of the prediction process so that the maps can be rapidly updated as new ground-truth data is obtained. Our modeling follows three phases:

(a) Model selection: we compare possible models of interest for PNV mapping and choose the optimal spatial prediction framework based on the cross-validation results.
(b) Model assessment: we assess the uncertainty of predictions per vegetation class and try to determine objectively the limitations of the mapping products for wider uses.
(c) Prediction: we use the best performing models to produce spatial predictions, then visually assess maps and if necessary repeat steps a–c.

## MATERIALS AND METHODS

### Theory

Potential natural vegetation is the hypothetical vegetation cover that would be present if the vegetation were in equilibrium with environmental controls, including climatic factors and disturbance, and not subject to human management.
When considering PNV, one needs to distinguish between potential *"natural"* and potential *"managed"* vegetation, and *"actual"* natural and *"actual"* managed vegetation (Fig. 1A). Vegetation is in general a dynamic feature. Also PNV changes as the climatic conditions change. For example, with the future global warming and changes in our climate, PNV might be significantly different than pre-industrial revolution. Therefore it is important to reference PNV to the time period of interest, so that historic PNV and current or future PNV maps can be produced (Fig. 1B).

In addition to the differentiation between the potential and actual natural vegetation, there are also three sub-types of the PNV that need to be considered:

1. PNV model A: based on the autochthonous or native vegetation and living species only.
2. PNV model B: based on the autochthonous or native vegetation that includes also extinct species.
3. PNV model C: PNV based on any vegetation whether native or introduced or extinct.

Derivation of maps of PNV model A could be of interest to, for example, nature conservationists; PNV model C could be of more interest to, for example, forestry and agroforestry organizations as it provides an objective basis for introducing non-native species to a new area.

Conveniently, locations that have not been subject to human disturbance/management can provide relevant information about vegetation cover in historic times, which can

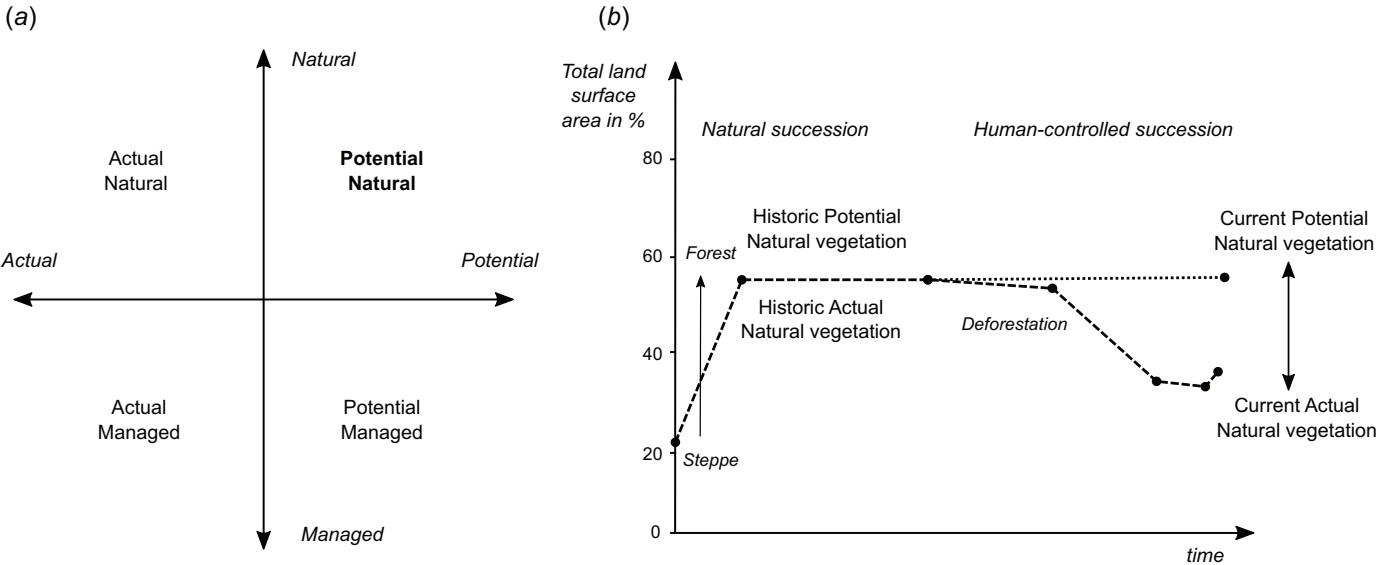

**Figure 1** Schematic explanation of differences between (A) potential and actual natural/managed vegetation, and (B) current and historic vegetation in the context of global land area.

serve as a guide to PNV. A major limitation of modeling PNV is that we unfortunately do not have equally detailed information about the status of vegetation and environment across historic periods. For instance, about half of the Earth's mature tropical forests have disappeared in the last 150 years and original habitats have been reduced to 10% (*Hansen et al., 2013*). Given that climates have changed and few areas are truly human impact *"free,"* even undisturbed historic vegetation only represents one possible expression of PNV for a given set of climate conditions at a specific time.

Regardless of the hypothetical nature of PNV, the concept (both as a classification and as a regression type problem) is still a helpful yardstick against which land cover change can be quantitatively measured and land restoration designs can be planned. *Erb et al. (2017)* have estimated that almost half of the standing global vegetation biomass carbon stocks has been lost, almost equally due to land cover change (e.g., tree cover to cropland) and management effects within land cover types (e.g., croplands managed at lower biomass carbon stocks than tree covered areas). PNV maps can thus help quantify such differences, both deficit and surplus, in biomass stocks caused by the current land management system more objectively and served as an input to the redesign of land management systems.

## PNV mapping and species distribution modeling

In principle, PNV mapping is a special case of species distribution modeling (*Elith & Leathwick, 2009*; *Hemsing & Bryn, 2012*; *Hijmans & Elith, 2018*): at the core of PNV mapping is statistical modeling of the relationship between species (or natural associations of species or communities) and a list of predictors, that is, biotic and abiotic site factors (*Elith & Leathwick, 2009*). The difference between mapping actual distribution

of species and PNV mapping is that PNV involves extrapolating the model to the whole land mask, assuming a hypothetical distribution under a specific set of undisturbed bioclimatic and/or biophysical conditions:

$$\Pr(Y) = f(\text{Relief}, \text{BioClimate}, \text{Lithology}) \tag{1}$$

where $Y$ is the target variable, which could be vegetation types or plant species with a finite number of states $Y \in \{1, 2, \ldots, k\}$ and/or vegetation properties. PNV mapping can be considered as a *classification-type* or *regression-type* problem depending on whether we map factors such as vegetation types or continuous vegetation properties such as biomass or leaf area index.

The primary assumptions we make when applying a PNV model to the training data are:

1. The ecological gradients captured in training data reflect only natural ecological gradients and not human controls such as land use systems, civil engineering constructions, or one-off major disturbance events such as volcanic eruptions, floods, or tsunamis.
2. Remote sensing data such NDVI often reflect human-altered vegetation patterns and ought not be used as covariates in PNV mapping (*Leong & Roderick, 2015*).
3. The training data are representative of the study area, especially considering the feature space (ecological gradients) of the study area.

Assuming a log-linear relationship between ecological gradients and target variables, PNV classes can be modeled using a multinomial log-linear model:

$$f(k, i) = \beta_{0,k} + \beta_{1,k}x_{1,i} + \beta_{2,k}x_{2,i} + \cdots + \beta_{M,k}x_{M,i} \tag{2}$$

where $f(k, i)$ is the linear predictor function, $\beta$ are the regression coefficients associated with the $m$th explanatory variable and the $k$th outcome. An efficient implementation of the multinomial logistic regression is the `multinom` function from the R package nnet (*Venables & Ripley, 2002*). The output of predictions produced using `multinom` are $k$ probability maps (0–100%) such that all predictions at each site sum up to 1:

$$\sum_{k=1}^{K} \Pr(Y_i = k) = 1 \tag{3}$$

In this paper, all prediction models are used in the *"probability"* mode; that is, to derive probability maps per class.

Note that a PNV spatial prediction model divides geographic space among all possible states given the training points. It is therefore necessary, for Eq. (1), that all possible states of $Y$ are represented with training data so that the model can be applied over the whole spatial domain of interest. If all of the states are not known, then the space will be artificially filled-in with known classes occupying similar ecological niches and which can lead to prediction bias. In other words, as with species distribution modeling of individual species, both presence and absence data play an equally important role for model calibration (*Elith & Leathwick, 2009*).

 

### Input data: training points

We consider three ground-truth data sets for model calibration:

1. An expanded version of the BIOME 6000 DB data set representing site-based reconstructions from surface pollen samples of major vegetation types or biomes (http://dx.doi.org/10.17864/1947.99).
2. EU Forest (*Mauri, Strona & San-Miguel-Ayanz, 2017*) and Global Biodiversity Information Facilities (GBIF) occurrence records of the 76 main forest tree taxa in Europe (http://dx.doi.org/10.15468/dl.fhucwx).
3. Long-term fraction of absorbed photosynthetically active radiation (FAPAR) monthly images derived using a time-series of Copernicus Global Land products (https://land.copernicus.eu).

BIOME 6000 and EU Forest and GBIF occurrences are point data sets, while FAPAR consists of remote sensing images at relatively fine spatial resolution (250 m), from which we sample a large number of values (ca 100,000) using random sampling after masking for areas of natural vegetation.

### BIOME 6000

The BIOME 6000 data set (http://dx.doi.org/10.17864/1947.99) includes vegetation reconstructions from modern pollen samples, preserved in lake and bog sediments and from moss polsters, soil and other surface deposits. The use of pollen data to reconstruct PNV relies on the fact that although modern pollen samples may contain markers of land use, the predominant pollen types found in any one sample are those of the regional vegetation within a radius on the order of 10–30 km around the sampling site. Even if forests have fragmented, these fragments continue to produce and disperse pollen grains, and the composition of the pollen assemblage provides information on tree taxa that are still present.

The BIOME 6000 data set is an amalgamation of multiple data sets. BIOME 6000 initially produced maps for individual regions: Europe, Africa and the Arabian Peninsula, the Former Soviet Union and Mongolia and China. Additional regions were subsequently added including Beringia, western North America, Canada and the eastern United States and Japan, and the data for northern Eurasia, China and southern Europe and Africa were also updated. These regional compilations were summarized in *Prentice & Jolly (2000)*. Subsequent regional updates include China (*Harrison et al., 2001*), the circum-Artic region (*Bigelow et al., 2003*), Australia (*Pickett et al., 2004*), and South America (*Marchant et al., 2009*). Additionally, we have also combined these data with pollen-based vegetation reconstructions from the Eastern Mediterranean-Black Sea-Caspian Corridor region (*Marinova et al., 2018*) available from http://dx.doi.org/10.17864/1947.109, to produce a more complete and up-to-date compilation of the BIOME 6000.

Some sites in the BIOME 6000 data set have multiple reconstructions based on multiple nearby modern pollen samples (up to 30), which provides a useful measure of the reconstruction uncertainty, but could lead to modeling bias because the number of modern

**Table 1 Summary results of cross-validation for mapping global distribution of biomes (20 classes).**

| Biome class | ME | TPR | AUC | N |
|---|---|---|---|---|
| Cold deciduous forest | −0.01 | 0.89 | 0.96 | 201 |
| Cold evergreen needleleaf forest | 0.01 | 0.87 | 0.98 | 892 |
| Cool evergreen needleleaf forest | −0.07 | 0.87 | 0.93 | 201 |
| Cool mixed forest | 0.01 | 0.86 | 0.97 | 1,549 |
| Cool temperate rain forest | 0.01 | 0.92 | 0.99 | 95 |
| Desert | 0.00 | 0.89 | 0.96 | 330 |
| Erect dwarf shrub tundra | −0.01 | 0.89 | 0.98 | 145 |
| Graminoid and forb tundra | −0.03 | 0.83 | 0.91 | 128 |
| Low and high shrub tundra | −0.01 | 0.88 | 0.98 | 393 |
| Prostrate dwarf shrub tundra | −0.02 | **0.54** | 0.90 | 11 |
| Steppe | 0.01 | 0.87 | 0.94 | 889 |
| Temperate deciduous broadleaf forest | −0.01 | 0.84 | 0.94 | 961 |
| Temperate evergreen needleleaf open woodland | 0.01 | 0.92 | 0.97 | 307 |
| Temperate sclerophyll woodland and shrubland | 0.00 | 0.94 | 0.99 | 154 |
| Tropical deciduous broadleaf forest and woodland | 0.01 | 0.86 | 0.97 | 215 |
| Tropical evergreen broadleaf forest | 0.00 | 0.87 | 0.99 | 333 |
| Tropical savanna | 0.01 | 0.89 | 0.99 | 291 |
| Tropical semi evergreen broadleaf forest | −0.05 | 0.87 | 0.98 | 160 |
| Warm temperate evergreen and mixed forest | 0.01 | 0.85 | 0.96 | 985 |
| Xerophytic woods scrub | −0.02 | 0.88 | 0.95 | 388 |

Note:
Classification accuracy for predicted class probabilities is based on fivefold cross-validation with refitting. ME = "mean error," TPR = "true positive rate," AUC = "area under curve," N = "number of occurrences."
Numbers in bold indicate critically low prediction accuracy.

samples varies between sites. To reduce these unwanted effects, we use only the most frequently reconstructed biome at each site and for those sites with two equally common reconstructions (ca. 900) we use both observations.

The number of biomes differentiated varies from region to region, and some biomes were only reconstructed in specific regions where they are particularly characteristic, although they may occur, but not be recognized, elsewhere. Furthermore, some biomes that can be recognized on the modern landscape were never reconstructed in the BIOME 6000 data set (e.g., cushion forb tundra)—either because of the sample distribution or because the characteristic plant-functional types were also spread amongst other biomes. Simplified or "megabiome" classifications (*Harrison & Bartlein, 2012*) involve a substantial loss of information. We have therefore created a new standardization of the classification scheme (see further Table 1; the final scheme has 20 globally applicable and distinctive biomes) which preserve the maximum number of distinct biomes that were reconstructed as present in multiple regions.

There are relatively few data vegetation reconstructions for tropical South America, which could lead to extrapolation problems and omission of important PNV classes in Latin America, but also potentially in tropical parts of Africa and Asia. To reduce under-representation of tropics, we have added 350 randomly simulated points based on
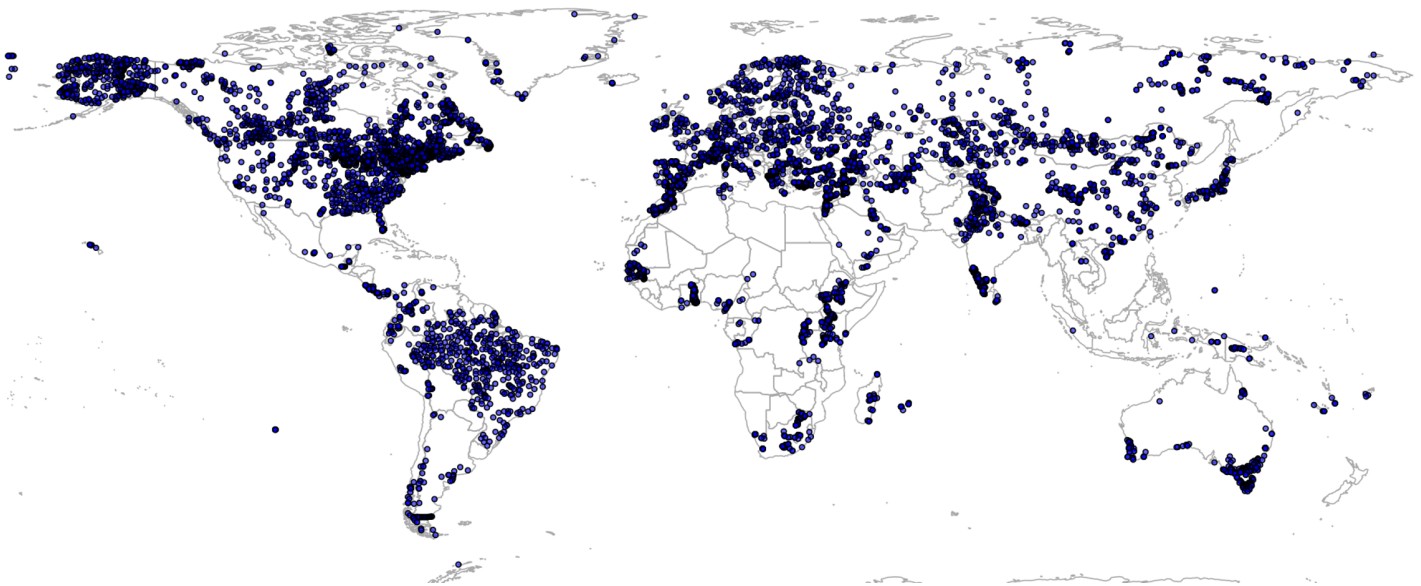

**Figure 2** **Spatial distribution of BIOME 6000 training points.** A total of 8,057 unique locations are shown on the map.

the RADAM Brazil natural vegetation polygon map at high spatial detail (Radam Vegetação SIRGAS map) (*Veloso et al., 1992*) obtained from ftp://geoftp.ibge.gov.br/. Before generating the pseudo-observations for Brazil, we translated SIRGAS map legends to match the BIOME 6000 classes. This translation is also available via the project's github repository. This gave a total of 8,057 unique individual locations represented in the combined data set, that is, a total of 8,959 training observations (Fig. 2).

We have mapped the distribution of biomes for all land pixels, with the exception of water bodies, barren land and permanent ice areas. Barren land and permanent ice areas were masked out using the ESA's global land cover maps for the period 2000–2015 (https://www.esa-landcover-cci.org) and the long-term FAPAR images, both available at relatively fine resolution of 300 m. We only mask out pixels that are permanent ice/barren ground and have a FAPAR = 0 throughout the period 2000–2015.

## European forest tree occurrence records

For mapping PNV distribution of forest tree taxa (note: most of these are individual species, but some are only recognized at sub-genus or genus level) in Europe we have merged two point data sets: EU Forest (*Mauri, Strona & San-Miguel-Ayanz, 2017*) (588,983 records covering 242 species) and GBIF occurrence records of the main forest tree taxa in Europe. The GBIF Occurrence data was downloaded on January 23, 2017 (http://dx.doi.org/10.15468/dl.fhucwx). We focus on modeling just the 76 forest tree taxa indicated in the European Atlas of Forest Tree Species (*San-Miguel-Ayanz et al., 2016*).

Global GBIF occurrence data can be obtained by using the rgbif package, in which case the only important parameter is the taxonKey (e.g., *"Betula spp."* corresponds to GBIF taxon key 2875008). After the bulk data download (which gives about four million occurrences), we imported all points and then subset occurrences based on the list of taxon

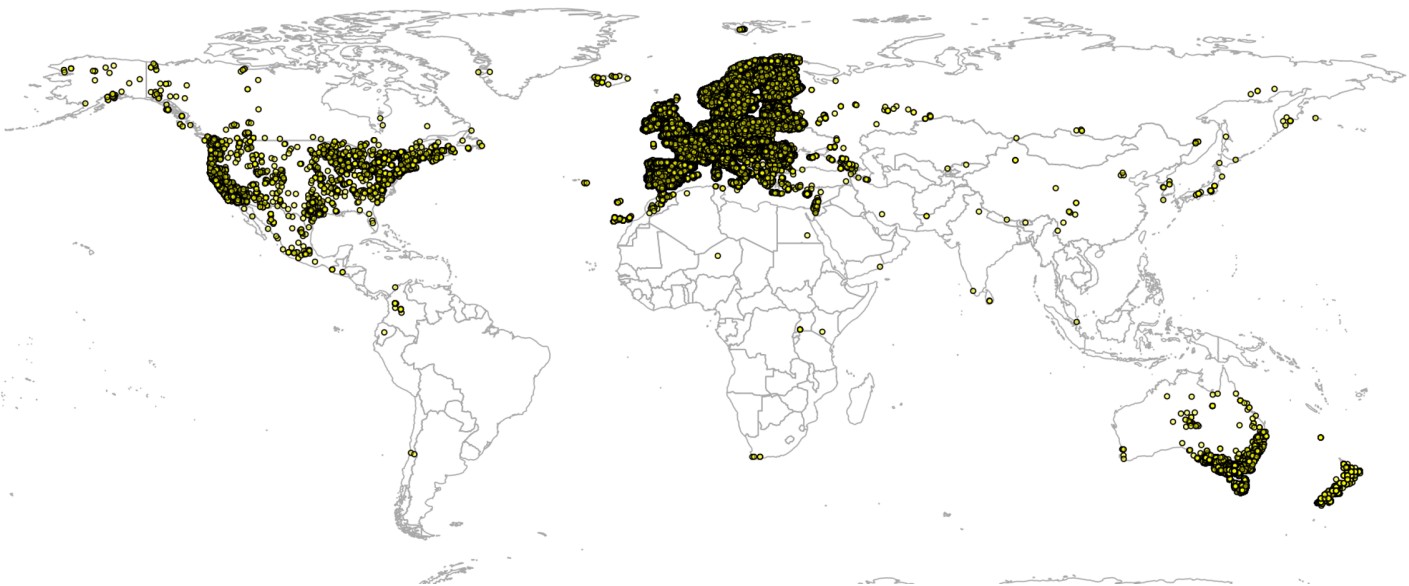

**Figure 3** Merge of EU Forest (*Mauri, Strona & San-Miguel-Ayanz, 2017*) and GBIF occurrence records used to build models to predict PNV for the 76 forest tree taxa. Total of 1,546,435 shown on the map.

keys and coordinate uncertainty (<2 km positional error). This gave a total of 1,546,435 training points from which about two thirds are GBIF points (Fig. 3). We assume in further analysis that the EU Forest point locations and representativeness are more trustworthy, hence we assign four times higher weights to these points than to the GBIF points.

Certain forest tree species (*Chamaecyparis lawsoniana*, *Eucalyptus globulus*, and *Pseudotsuga menziesii*), that are shown in the European Atlas of Forest Tree Species are introduced, that is, planted and do not generally propagate naturally. Hence, they were removed from the list of target forest tree species. We retained, however, three species (*Ailnthus altissima*, *Picea sitchensis*, and *Robinia pseudoacacia*) that are not native but are extensively naturalized. The total number of target forest tree taxa was 73.

We built predictive models for European forest tree taxa using information on their global distribution, but only generate predictions for Europe. In other words, we use a global compilation for model training to increase the precision of the definition of the ecological niche of each taxon, but then predict only for Europe as the selection of taxa is based on the European Atlas of Forest Tree Species (*San-Miguel-Ayanz et al., 2016*).

**Fraction of absorbed photosynthetically active radiation**

Fraction of absorbed photosynthetically active radiation monthly images for 2014–2017 were obtained from https://land.copernicus.eu (original values reported in the range 0–235 with scaling factor 1/255, i.e., with a maximum value of 0.94). From a total of 142 images downloaded from https://land.copernicus.eu we derived minimum, median and maximum value of FAPAR per month (12) using the 95% probability interval using the data.table package (http://r-datatable.com). For regression modeling we only report results of predictions of median values of FAPAR; predictions of minimum and maximum FAPAR can be obtained from the data repository.

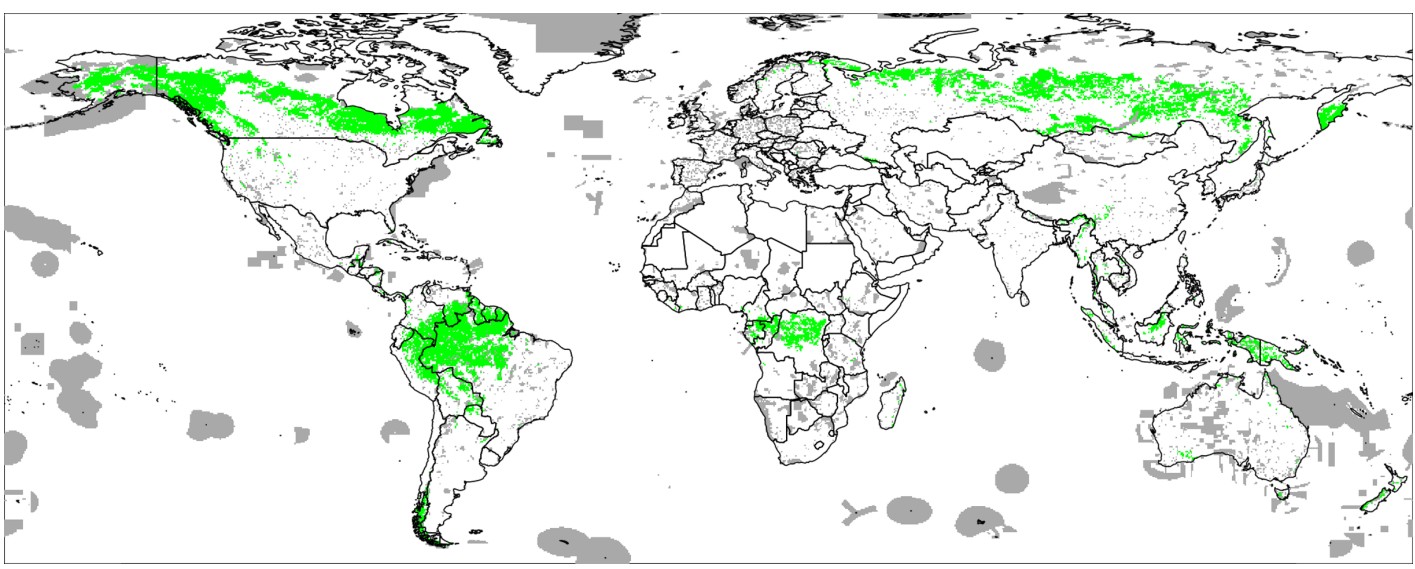

**Figure 4 World's Protected Areas (dark gray) based on http://protectedplanet.net and Intact Forest Landscapes for year 2000 (green) based on http://intactforests.org.** These maps were used to randomly select some 30,000 training points to predict potential FAPAR under PNV.

We model median and upper 95% FAPAR values as a function of the same covariate layers used in all three case studies. For model training we use ca. 30,000 randomly sampled points (simple random sampling) exclusively from protected area as shown in the World Database on Protected Areas (WDPA) data set (http://protectedplanet.net) and the intact forest landscapes (IFL) data set for 2000 and 2013 (*Potapov et al., 2008*; Fig. 4). We use about three times more training points from the IFL 2013 areas for model development than from the WDPA and IFL 2000 masks to emphasize more ecological conditions of intact vegetation.

The prediction model for FAPAR under PNV is in the form of:

`R > FAPAR ~ cm + X1m + X2m + X3 + ... + Xp`

where `X1m` is the covariate with monthly values (e.g., precipitation, day-time and night-time temperatures, etc.), `X3` is the environmental covariates that do not vary through year (e.g., lithology or DEM derivatives), and $c_m$ is the cosine of the month number:

$$c_m = \cos(\mu/12 \cdot 2 \cdot \pi) \tag{4}$$

where $\mu$ is the month number 1–12. The total number of training observations used to build models is in fact 180,483 (each training site is represented up to 12 times).

For PNV FAPAR mapping we have masked out all water bodies including lakes and rivers, following the ESA's global land cover maps for the period 2000–2015 (https://www.esa-landcover-cci.org) and permanent ice/barren ground.

### Input data: environmental covariates

For modeling purposes, we use a stack of 160 spatially explicit co-variate data layers that represent standard ecological gradients essential for growth and survival of plants:

- DEM derivatives quantifying various landscape metrics and hydrological processes: slope, curvature, topographic index, topographic openness, valley depth, and multi-resolution valley bottom index; all derived using the SAGA GIS (*Conrad et al., 2015*).
- Mean, minimum and maximum monthly temperatures derived as a mean between WorldClim v2 (http://worldclim.org/version2) and CHELSA climate (*Karger et al., 2017*).
- Mean monthly precipitation images derived as a weighted average between the WorldClim v2, CHELSA climate and Global Precipitation Measurement Integrated Multi-satellitE Retrievals for GPM rainfall product.
- CHELSA Bioclimatic layers downloaded from http://chelsa-climate.org/, including: annual mean temperature, mean diurnal temperature range, isothermality (day-to-night temperature oscillations relative to the summer-to-winter oscillations), temperature seasonality (standard deviation (s.d.) of monthly temperature averages), maximum temperature of warmest month, minimum temperature of coldest month, temperature annual range, mean temperature of warmest quarter, mean temperature of coldest quarter, annual precipitation amount, precipitation of wettest month, precipitation of driest month, precipitation of wettest quarter, precipitation of driest quarter (*Karger et al., 2017*).
- European Space Agency's CCI-LC snow probability monthly averages based on MODIS snow products MOD10A2 downloaded from http://maps.elie.ucl.ac.be/CCI/viewer/index.php.
- USGS Global Ecophysiography landform classification and lithological map at 250 m resolution obtained from http://rmgsc.cr.usgs.gov/outgoing/ecosystems/Global/ and based on Global Lithological Map (GLiM) (*Hartmann & Moosdorf, 2012*).
- MODIS Cloud fraction monthly images obtained from http://www.earthenv.org/cloud (*Wilson & Jetz, 2016*).
- Global Water Table Depth in meters based on *Fan, Li & Miguez-Macho (2013)*.
- NASA's monthly MODIS Precipitable Water Vapor images (MYDAL2_M_SKY_WV data set at http://neo.sci.gsfc.nasa.gov).
- Potential wetlands GIEMS map (*Fluet-Chouinard et al., 2015*).
- Global Surface Water dynamics images: occurrence probability, surface water change, and water maximum extent; downloaded from https://global-surface-water.appspot.com/download (*Pekel et al., 2016*).
- Density of earthquakes based on the USGS Earthquake Archives (http://earthquake.usgs.gov/earthquakes/).

Some CHELSA bioclimatic layers contained too many missing pixels or artifacts (e.g., mean temperature of wettest quarter, mean temperature of driest quarter, precipitation seasonality, precipitation of warmest quarter, and precipitation of coldest quarter) and hence were not used for further modeling to avoid propagating those artifacts to final predictions.

All original layers have been resampled to the standard grid at a spatial resolution of 1/120 decimal degrees (about one km) covering latitudes between −62.0 and 87.37. Some layers such as water vapor needed to be downscaled from 10 to 1 km resolution, for which we used the bicubic splines algorithm as implemented in GDAL (*Mitchell & GDAL Developers, 2014*). We do not map Antarctica as this continent is dominantly covered with permanent ice and there are no training points. We limit all analysis to one km, that is, 1/120 degrees in geographical coordinates, to avoid too high of a computational load, even though many of environmental covariates are also available at finer resolutions.

We use the same stack of covariates for mapping global distribution of biomes, FAPAR and forest tree species in Europe, in order to be able to compare model performance and investigate whether the most important covariates differ among the three case studies.

## Machine learning algorithms examined

We examine predictive performance of the following machine learning algorithms (MLAs):

- Neural networks (*Venables & Ripley, 2002*).
- Random forest (RF; *Breiman, 2001*; *Cutler et al., 2007*; *Biau & Scornet, 2016*; *Hengl et al., 2018*).
- Generalized boosted regression models (*Friedman, 2002*).
- K-nearest neighbors (*Venables & Ripley, 2002*).

Neural networks are available from several packages in R. Here, we use the nnet package (*Ripley & Venables, 2017*) also described in *Venables & Ripley (2002)*. RF is efficiently implemented in the ranger package (*Wright & Ziegler, 2017*) and can be used to process large data sets. Generalized boosted regression models are available via the gbm package (*Ridgeway, 2017*). The K-nearest neighbor regression is available via the class package, that is, the knn function (*Venables & Ripley, 2002*). Of these four algorithms, the K-nearest neighbors is computationally the least intensive and results in relatively simple models, while RF is computationally the most intensive and results in large models. However, a limitation of the K-nearest neighbors approach is that it does not handle high dimensional data in comparison to RF or neural nets.

We also test using the same packages to fit models for regression-type problems (e.g., modeling of FAPAR), with the exception of the class package, that is, the `knn` function which can only be used for classification problems. For modeling FAPAR we instead added use of the Cubist approach, available via the Cubist package (*Kuhn et al., 2017*), and the extreme gradient boosting approach available via the xgboost package (*Chen & Guestrin, 2016*).

The caret package has many more MLA of interest for classification and regression problems than presented here, but many are not fully optimized for large data sets and hence also not applicable for large data sets (>> 1,000 observations with >> 100 covariates).
## Model selection

For model fitting and model selection we use the caret package implementation for automated evaluation of models. When comparing performance of the models we look at classification accuracy based on cross-validation with refitting implemented in the `caret` package via the setting (*Kuhn, 2008*; *Kuhn & Johnson, 2013*):

```
R > ctrl < -trainControl(method = "repeatedcv", number = 5, repeats = 2)
```

which translates as: models are refit five times using 80% of the data and predictions derived from the fitted models are compared with the remaining observations; this process is then repeated two times to produce stable results. The reported accuracy is the map accuracy (0–100%) and/or root mean square error (RMSE) derived using all merged cross-validations (*Kuhn, 2008*; *Kuhn & Johnson, 2013*). Since most of the data sets are fairly large and model fitting can take hours, even in a high performance computing environment, we limit the number of repetitions to 2.

For FAPAR (regression modeling) and selection of the final prediction model we use the same repeated cross-validation as implemented via the `caret` package. This is, in principle, similar to evaluation of the classification accuracy, except the comparison criterion is RMSE.

All analyses were run on a high performance computing Amazon ec2 server with 64 threads (32 CPU's) and 256 GiB RAM. Total computing time to produce all outputs is about 12 hours of optimized computing (or about 600 CPU hours). One kilometer data can be processed with two degree tiles, which usually requires some 5,000 tiles to represent the land mask. All processing steps and preparation of input and output maps are fully documented at https://github.com/envirometrix/PNVmaps. All output maps are available for download via http://dx.doi.org/10.7910/DVN/QQHCIK under the Open Database License.

## Performance of classification algorithms

Performance of classification algorithms is assessed using fivefold cross-validation with refitting of models. For evaluation of the mapping accuracy for biomes and tree species we use the map purity (0–100%) and kappa metrics for the dominant (hard) classes as the key measures of predictive performance (*Kuhn & Johnson, 2013*). For each class we also provide predicted probabilities, which can be used to model transition zones and correlation between classes. For the predicted probabilities of class occurrences (0–1) we derived the true positive rate (TPR) and the area under the receiver operating characteristic curve (AUC) as implemented in the ROCR package (*Sing et al., 2005*, *2016*). TPR value = 1 indicates a perfect match to the class positives in ground data while TPR values <0.5 can be considered poor mapping accuracy. Likewise, values of AUC close to 1 indicate high prediction performance, while values around 0.5 and below are considered poor. TPR and AUC provide probably a more informative measure of the mapping accuracy than overall mapping accuracy/kappa, as they also allow detection of problematic classes.

We also use Scaled Shannon Entropy Index (SSEI), which can be derived using the per-class probability maps (*Shannon, 1949*; *Borda, 2011*):

$$\text{SSEI}_s(x) = -\sum_{i=1}^{b} P_i(x) \cdot \log_b P_i(x) = \frac{-\sum_{i=1}^{b} P_i(x) \cdot \log P_i(x)}{-b \cdot b^{-1} \cdot \log b^{-1}} \tag{5}$$

where $b$ is the total number of possible classes and $P$ is probability of class $i$. The SSEI is in the range from 0–1, where 0 indicates a perfect classification and 1 (or 100%) indicates maximum confusion. SSEI should not be confused with classification accuracy assessment. For example, $\text{SSEI}_s$ <60% indicates relatively low confusion between classes, that is, high accuracy, while mapping error of 60% would be considered a relatively poor classification accuracy result.

For the biomes data set, where spatial clustering of points is significant, we also use repeated spatial cross-validation as implemented in the mlr package (*Bischl et al., 2016*):

```
R > learner.rf = makeLearner("classif.ranger", predict.type = "prob")
R > resampling = makeResampleDesc("SpRepCV", fold = 5, reps = 5)
```

It has been shown that spatial autocorrelation in data and serious spatial clustering in training points can lead to somewhat biased estimate of the actual accuracy (*Brenning, 2012*). A solution to this problem is to apply spatial partitioning so that possible bias due to spatial proximity is minimized.

We also compare results of modeling potential distribution of tree species in Europe with the habitat type maps of Europe produced independently by *San-Miguel-Ayanz et al. (2016)* and *Brus et al. (2012)*. This comparison is visually based only.

## Performance of regression algorithms

Performance of regression algorithms is also assessed using fivefold cross-validation with refitting of models. For assessment of the mapping accuracy for FAPAR we use as the main performance measures the RMSE:

$$\text{RMSE} = \sqrt{\frac{\sum_{j=1}^{m} \left[\hat{y}(\mathbf{s}_j) - y(\mathbf{s}_j)\right]^2}{n}} \tag{6}$$

and mean error (ME):

$$\text{ME} = \frac{\sum_{j=1}^{m} \left[\hat{y}(\mathbf{s}_j) - y(\mathbf{s}_j)\right]}{n} \tag{7}$$

where $\hat{y}(\mathbf{s}_j)$ is the predicted value of $y$ at the cross-validation location, and $m$ is total number of cross-validation points. We also report amount of variation explained by the model ($R^2$) derived as:

$$R^2 = \left[1 - \frac{\text{SSE}}{\text{SST}}\right] \times 100\% \tag{8}$$

where SSE is the sum of squared errors at cross-validation points and SST is the total sum of squares. A coefficient of determination close to 1 indicates a perfect model.

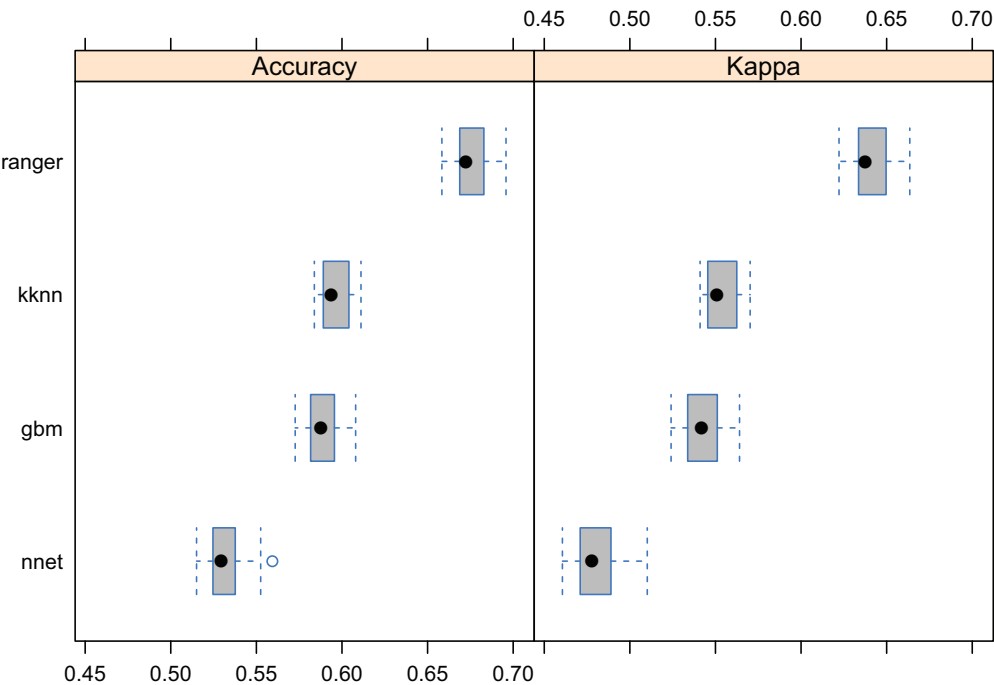

**Figure 5 Predictive performance of the target machine learning algorithms for mapping global distribution of biomes ($N$ = 8,653; spatial distribution of training points is available in Fig. 2).** Methods compared: ranger, random forest; kknn, K-nearest neighbors; gbm, generalized boosted regression models; nnet, neural networks.

# RESULTS

## Global maps of biomes

Results showed that a relatively accurate model of PNV could be produced from the BIOME 6000 data set using the existing stack of covariates at one km spatial resolution. Results of cross-validation show the RF model to be the best performing method and distinctively superior to all other approaches (Fig. 5). The choice of the RF mtry parameter had little impact on overall accuracy, most likely because there was a high overlap in covariate maps so that even with smaller `mtry` bagging the performance was relatively similar. The best prediction accuracy from among the four methods used for mapping global biomes was about 68%. The predicted biome classes are presented in Fig. 6.

The most important covariates for the RF model were: total annual precipitation, monthly temperatures, CHELSA bioclimatic layers, atmospheric water vapor images, and monthly precipitation. Landform parameters and lithology are not amongst the top 20 most important predictors. The decline in variable importance was, however, gradual—even lower ranked covariates might still affect the accuracy of predictions.

The detailed cross-validation results show that the only difficult class to predict was prostrate dwarf shrub tundra (Table 1). The TPR value for most class probabilities ranges from 0.83 to 0.94 indicating relatively high match with ground data. The SSEI map

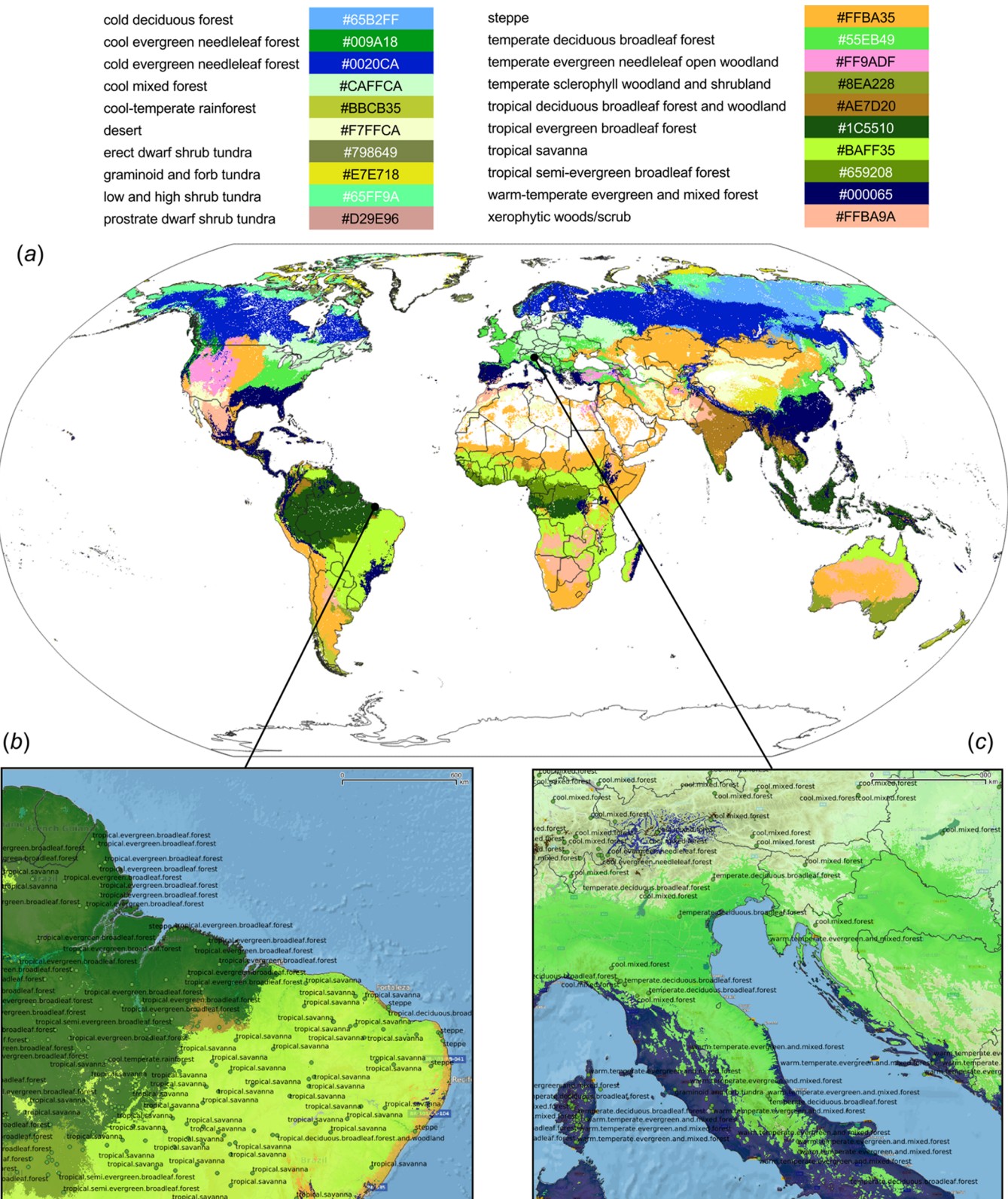

**Figure 6 Predicted PNV distribution for (A) global biomes with a zoom in on areas in Brazil (B) and Europe (C).** Labels indicates training points from the BIOME 6000 data set (Fig. 2). Background map data: Google, DigitalGlobe.

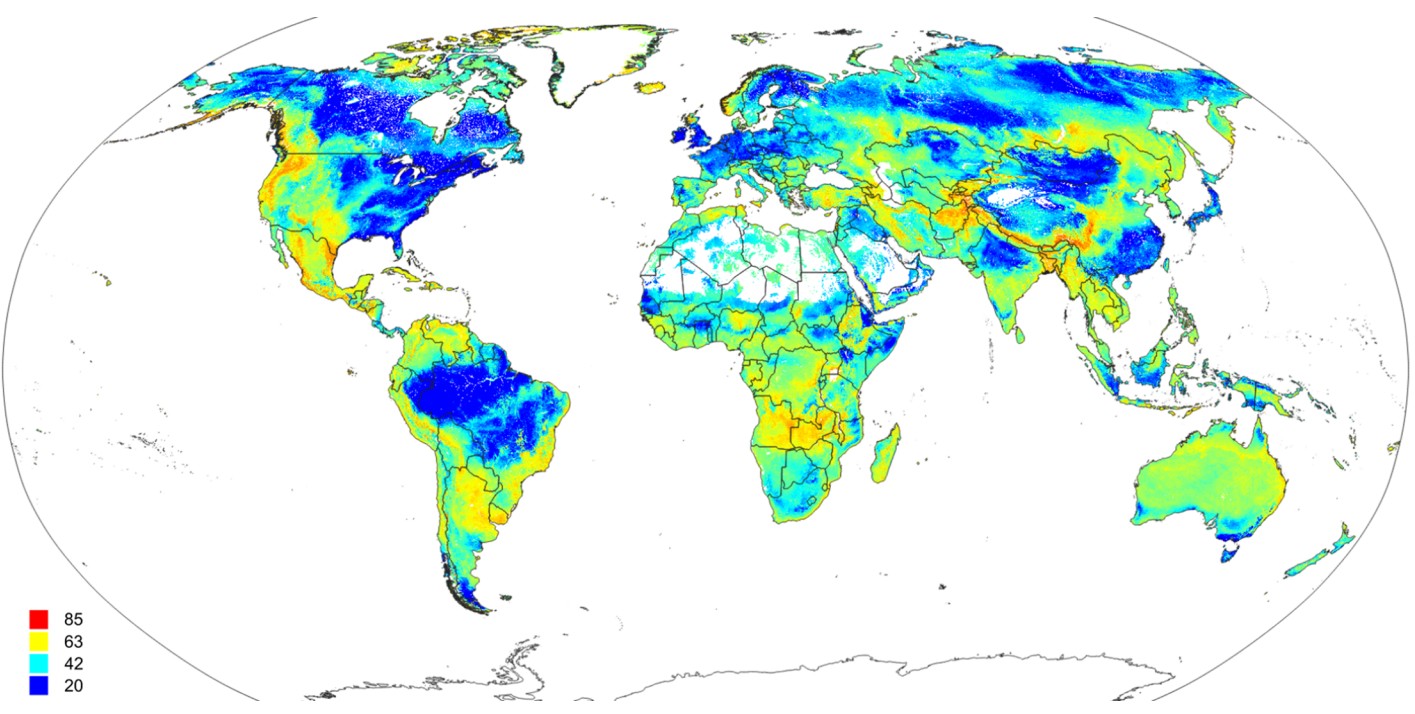

**Figure 7 Scaled Shannon Entropy Index (SSEI) derived using predicted probabilities for 20 biomes (classes) based on Eq. (5).** High values of SSEI (red color) indicate high confusion between classes.

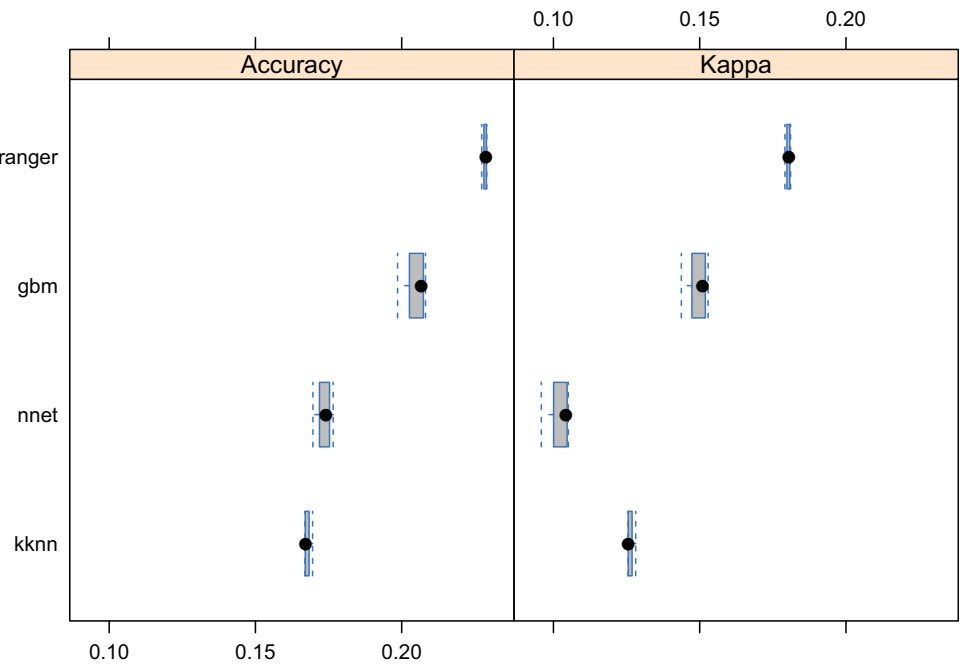

**Figure 8 Predictive performance of the target machine learning algorithms for mapping forest tree species (N = 1.5 million distribution of training points is available in Fig. 3).** Methods compared: ranger, random forest; gbm, generalized boosted regression models; nnet, neural networks; kknn, K-nearest neighbors.

**Table 2  Results of cross-validation for the forest tree taxa.**

| Species name | GBIF taxon ID | ME | TPR | AUC | N |
|---|---|---|---|---|---|
| *Abies alba* | 2685484 | −0.01 | 0.77 | 0.92 | 16,150 |
| *Acer campestre* | 3189863 | −0.01 | 0.65 | 0.83 | 19,819 |
| *Acer platanoides* | 3189846 | −0.02 | 0.68 | 0.82 | 30,801 |
| *Acer pseudoplatanus* | 3189870 | −0.01 | 0.69 | 0.79 | 65,039 |
| *Aesculus hippocastanum* | 3189815 | −0.01 | 0.59 | 0.85 | 8,088 |
| *Ailanthus altissima* | 3190653 | 0.04 | 0.69 | 0.92 | 1,576 |
| *Alnus cordata* | 2876607 | 0.05 | 0.73 | 0.95 | 904 |
| *Alnus glutinosa* | 2876213 | 0.00 | 0.71 | 0.77 | 91,292 |
| *Alnus incana* | 2876388 | −0.03 | 0.76 | 0.95 | 6,873 |
| *Betula* spp. | 2875008 | −0.03 | 0.63 | 0.83 | 7,313 |
| *Carpinus betulus* | 2875818 | 0.00 | 0.75 | 0.89 | 22,765 |
| *Carpinus orientalis* | 2875780 | 0.07 | **0.21** | 0.92 | 284 |
| *Castanea sativa* | 5333294 | 0.00 | 0.74 | 0.91 | 13,049 |
| *Celtis australis* | 2984492 | −0.01 | 0.54 | 0.92 | 594 |
| *Cornus mas* | 3082263 | 0.03 | 0.51 | 0.90 | 827 |
| *Cornus sanguinea* | 3082234 | −0.03 | 0.59 | 0.82 | 8,837 |
| *Corylus avellana* | 2875979 | −0.02 | 0.67 | 0.76 | 48,140 |
| *Cupressus sempervirens* | 2684030 | −0.04 | **0.21** | 0.70 | 284 |
| *Euonymus europaeus* | 3169131 | −0.02 | 0.61 | 0.83 | 12,119 |
| *Fagus sylvatica* | 2882316 | 0.00 | 0.73 | 0.81 | 89,044 |
| *Frangula alnus* | 3039454 | −0.02 | 0.71 | 0.86 | 26,873 |
| *Fraxinus angustifolia* | 7325877 | −0.05 | 0.63 | 0.94 | 1,757 |
| *Fraxinus excelsior* | 3172358 | 0.00 | 0.67 | 0.74 | 91,111 |
| *Fraxinus ornus* | 3172347 | 0.02 | 0.86 | 0.99 | 2,765 |
| *Ilex aquifolium* | 5414222 | −0.01 | 0.66 | 0.82 | 26,873 |
| *Juglans regia* | 3054368 | −0.03 | 0.60 | 0.89 | 3,643 |
| *Juniperus communis* | 2684709 | −0.03 | 0.71 | 0.86 | 21,189 |
| *Juniperus oxycedrus* | 2684451 | −0.07 | 0.71 | 0.97 | 1,705 |
| *Juniperus phoenicea* | 2684640 | −0.07 | 0.74 | 0.98 | 1,137 |
| *Juniperus thurifera* | 2684528 | −0.03 | 0.87 | 0.99 | 1,886 |
| *Larix decidua* | 2686212 | −0.01 | 0.71 | 0.89 | 15,581 |
| *Olea europaea* | 5415040 | 0.00 | 0.90 | 0.99 | 7,080 |
| *Ostrya carpinifolia* | 5332305 | 0.06 | 0.90 | 0.99 | 1,809 |
| *Picea abies* | 5284884 | 0.02 | 0.76 | 0.86 | 122,713 |
| *Picea sitchensis* | 5284827 | 0.05 | 0.80 | 0.96 | 13,023 |
| *Pinus cembra* | 5285134 | −0.01 | 0.77 | 0.96 | 853 |
| *Pinus halepensis* and<br>  *Pinus brutia* | 5285604 | 0.03 | 0.86 | 0.99 | 16,951 |
| *Pinus mugo* | 5285385 | 0.00 | 0.85 | 0.98 | 6,667 |
| *Pinus nigra* | 5284809 | 0.01 | 0.79 | 0.93 | 13,540 |
| *Pinus pinaster* | 5285565 | 0.01 | 0.86 | 0.98 | 17,080 |
| *Pinus pinea* | 5285165 | −0.04 | 0.85 | 0.99 | 4,910 |
| *Pinus sylvestris* | 5285637 | 0.02 | 0.78 | 0.85 | 153,928 |

| Species name | GBIF taxon ID | ME | TPR | AUC | N |
| --- | --- | --- | --- | --- | --- |
| *Populus alba* | 3040233 | −0.01 | 0.54 | 0.86 | 4,522 |
| *Populus nigra* | 3040227 | −0.01 | 0.65 | 0.89 | 5,478 |
| *Populus tremula* | 3040249 | −0.02 | 0.66 | 0.74 | 44,057 |
| *Prunus avium* | 3020791 | −0.01 | 0.63 | 0.77 | 25,711 |
| *Prunus cerasifera* | 3021730 | 0.00 | 0.73 | 0.94 | 3,928 |
| *Prunus mahaleb* | 3022789 | −0.01 | **0.31** | 0.75 | 517 |
| *Prunus padus* | 3021037 | −0.03 | 0.63 | 0.78 | 21,705 |
| *Prunus spinosa* | 3023221 | −0.01 | 0.69 | 0.81 | 31,783 |
| *Quercus cerris* | 2880580 | 0.00 | 0.80 | 0.97 | 4,109 |
| *Quercus ilex* | 2879098 | 0.02 | 0.85 | 0.99 | 22,972 |
| *Quercus pubescens* | 2881283 | 0.01 | 0.86 | 0.98 | 9,096 |
| *Quercus pyrenaica* | 2878826 | 0.00 | 0.88 | 0.99 | 6,253 |
| *Quercus robur* and *Quercus petraea* | 2878688 | 0.01 | 0.69 | 0.76 | 141,938 |
| *Quercus suber* | 2879411 | −0.04 | 0.86 | 0.99 | 5,504 |
| *Robinia pseudoacacia* | 5352251 | 0.01 | 0.71 | 0.90 | 13,411 |
| *Salix alba* | 5372513 | 0.02 | 0.72 | 0.90 | 11,938 |
| *Salix caprea* | 5372952 | −0.03 | 0.68 | 0.78 | 40,879 |
| *Sambucus nigra* | 2888728 | 0.00 | 0.70 | 0.81 | 44,961 |
| *Sorbus aria* | 3012680 | −0.01 | 0.59 | 0.87 | 5,426 |
| *Sorbus aucuparia* | 3012167 | −0.01 | 0.70 | 0.76 | 86,977 |
| *Sorbus domestica* | 3013206 | −0.04 | **0.48** | 0.87 | 801 |
| *Sorbus torminalis* | 3012567 | −0.03 | 0.62 | 0.92 | 2,558 |
| *Taxus baccata* | 5284517 | −0.02 | 0.58 | 0.82 | 8,062 |
| *Tilia* spp. | 3152041 | −0.02 | 0.50 | 0.82 | 4,393 |
| *Ulmus* spp. | 2984510 | −0.03 | 0.64 | 0.92 | 5,426 |
| *Tilia* spp. | 3152041 | 0.00 | 0.58 | 0.85 | 4,522 |
| *Ulmus* spp. | 2984510 | −0.02 | 0.69 | 0.91 | 5,375 |

**Note:**

Classification accuracy for predicted class probabilities based on fivefold cross-validation. ME = "mean error," TPR = "true positive rate," AUC = "area under curve," N = "Number of occurrences." Taxa with less than <50 observations were omitted from analysis.

Numbers in bold indicate critically low prediction accuracy.

(Fig. 7) showed that the zones of highest confusion between classes can be found in Afghanistan, Nepal, mountainous parts of the USA and Mexico, parts of Angola and Zambia. The map of the SSEI is comparable to the confusion map produced by *Levavasseur et al. (2012)*, except in our case the Rocky Mountains in USA and mountains chains in South America show somewhat higher confusion. Many of the areas with high confusion index occur because the prediction model has problems distinguishing between closely-related biomes such as the *"cold evergreen needleleaf forest"* and *"cool evergreen needleleaf forest"* (e.g., Scotland).

Results of the accuracy assessment based on the spatial cross-validation (mlr package implementation; *Bischl et al., 2016*) further indicate that the spatial clustering of points

(a)

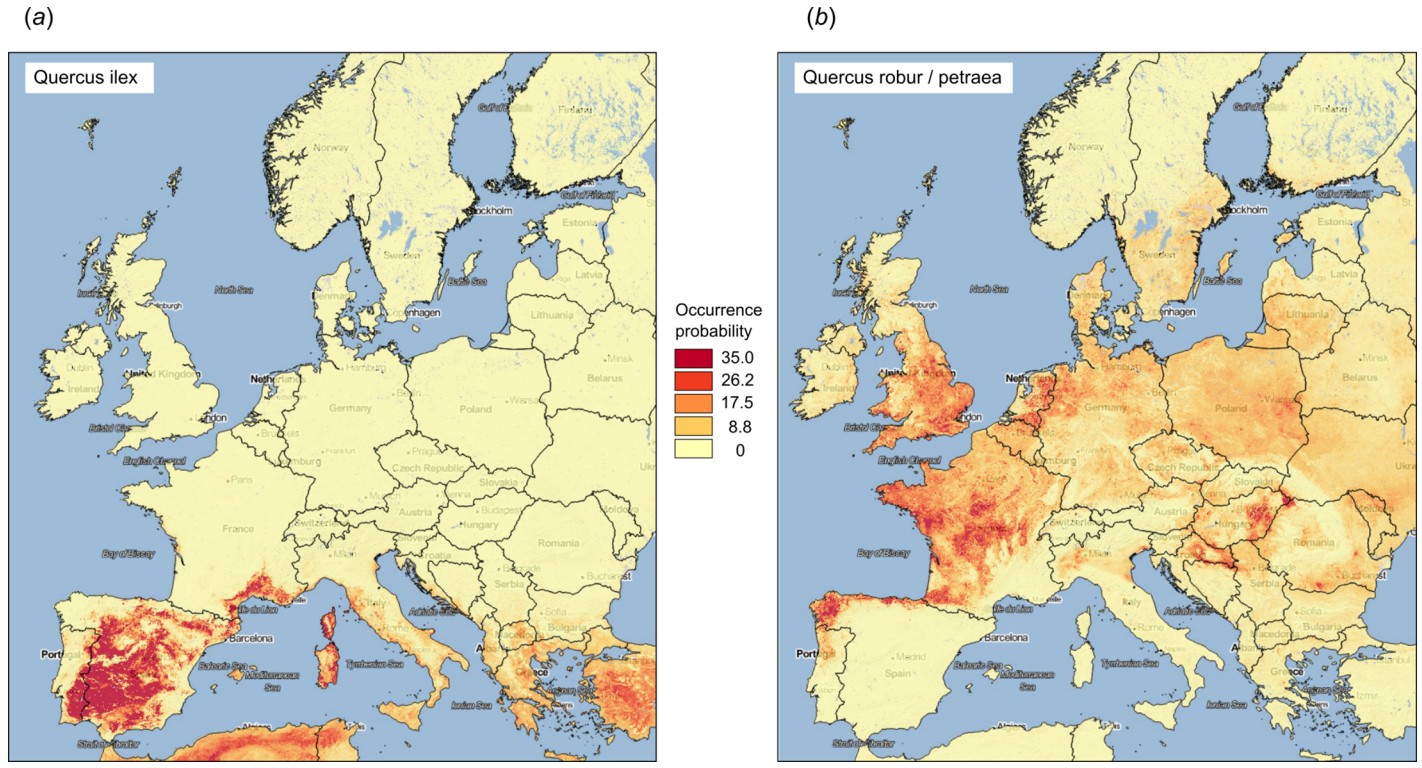

(b)

**Figure 9 Examples of predicted PNV distributions (probabilities) for European forest tree species (A)** *Quercus Ilex* **(GBIF ID: 2879098; 36,724 training points) and (B)** *Quercus robur/petraea* **(GBIF ID: 2878688; 404,296 training points).** Background map data: Google, DigitalGlobe.

does have a large effect on the mapping accuracy: spatial CV drops from 0.68 to 0.33 and weighted kappa to 0.45. This likely happens due to high spatial clustering of the biome points and due to the high spatial autocorrelation of biomes.

## European forest tree species

The results of fivefold cross-validation with re-fitting at each fold, confirms that RF was also the best prediction method for the forest taxa data set (Fig. 8). The overall mapping accuracy was significantly lower than for biomes, but this reduction in accuracy was to be expected as many of these taxa occur in communities, resulting in natural overlap of forest tree taxa distribution. The mapping accuracy of individual taxa, however, can be relatively high with TPR values of between 0.16–0.90 and an average value of around 0.69 (Table 2). The final maps (Fig. 9) showed a relatively good match with ground data, meaning that with the exception of some species of rarer occurrence (*Picea omorika, Cupressus sempervirens, Prunus mahaleb*), the species probability distribution maps were relatively accurate.

The most important predictors in the RF model for forest tree taxa were mean annual daily temperature, other monthly temperatures, elevation, CHELSA bioclimatic images, monthly precipitation, and MODIS cloud fraction images. Covariates for lithology and landform classification did not feature in the top 20 predictors. It could be that the GLiM (*Hartmann & Moosdorf, 2012*), which was used to represent changes in lithology, is too general for this scale of work.

(a)

(b)

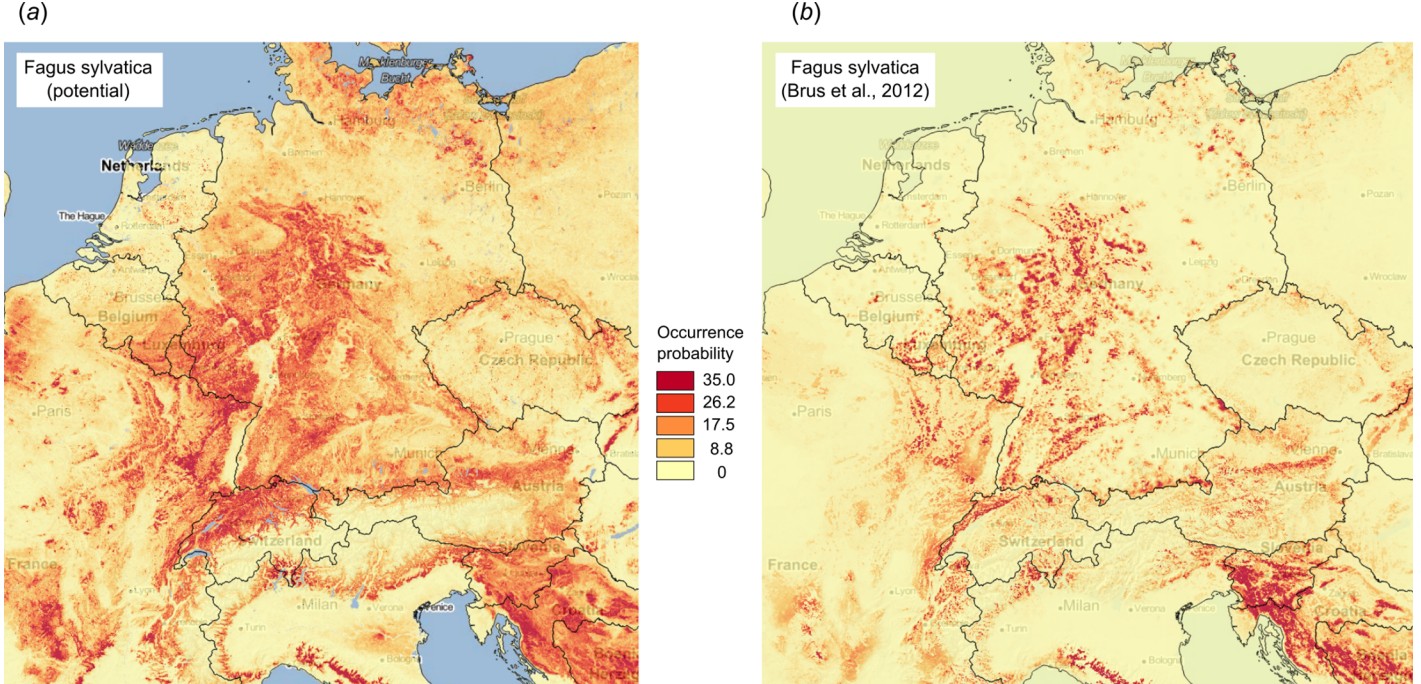

**Figure 10** Comparison between predicted PNV distribution for (A) *Fagus sylvatica* (GBIF ID: 2882316) based on our results, and (B) based on the maps generated by *Brus et al. (2012)*, that is, showing the presumed actual distribution of the tree species. Background map data: Google, DigitalGlobe.

Figure 10 illustrates differences between the map of actual distribution of *Fagus sylvatica*, generated by *Brus et al. (2012)*, and our predictions. In this case, the potential for extending habitat of *F. sylvatica* is significant, especially over parts of France and Germany.

Correlation analysis using all predicted distribution maps (matrix of Pearson's rho rank correlation coefficients for all possible pairs) indicated that many forest species are positively correlated, especially *F. sylvatica* and *Abies alba* and *Populus nigra* and *Salix alba*. High overlap between species probability maps reflects co-existence within communities, and thus could help with objectively defining forest communities.

## Global monthly FAPAR

The RF approach also produced the best preditcions of potential FAPAR (Fig. 11). The models for FAPAR were highly significant with R-squared around 90% and RMSE at ± 24 (original values in the range 0–232 where 235 corresponds to FAPAR = 100%) for the most accurate model based on fivefold Leave-Location-Out cross-validation. However, unlike with biomes and forest species distributions, the regression-tree Cubist model achieves equal performance to that of RF. The most important covariates for predicting FAPAR were total annual precipitation, MODIS cloud fraction images, CHELSA bioclimatic images, and monthly precipitation images. The `caret` package further suggested that mtry parameter for RF needs to be set higher than the default values for modeling FAPAR. Setting up mtry >25 helps reduce the RMSE by about 7–8%.

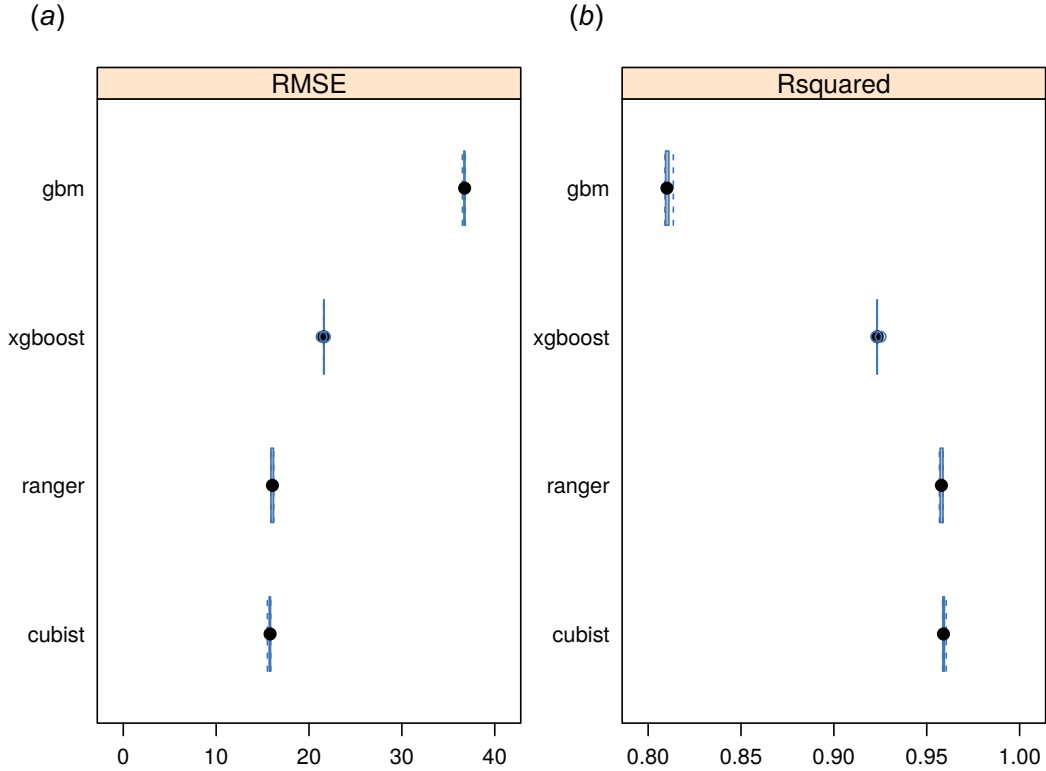

**Figure 11 Predictive performance of four machine learning algorithms for mapping global distribution of FAPAR ($N$ = 180,990).** Methods compared: gbm, generalized boosted regression models; xgboost, Extreme Gradient Boosting; ranger, random forest; cubist, Cubist regression models. (A) RMSE, root mean square error, (B) R-squared.               

Figure 12 depicts an example of actual vs predicted (PNV based) FAPAR for February in the urban area around São Paulo, where lower actual FAPAR reflects the removal of natural vegetation. Even larger differences between the potential and actual FAPAR are observed in parts of Africa (Fig. 13), likely reflecting land degradation and destruction of vegetation cover. In areas of intensive agricultural production (e.g., Western Australia and Midwest USA), actual FAPAR can be much higher than potential FAPAR under potential natural vegetation in a given month. However, this is often a temporal effect, as when PNV FAPAR is aggregated over the whole year, most places modified by human management show actual FAPAR is lower than potential. In Western Australian cropping zones, for example, crop fields have higher FAPAR during the winter growing season, but since the fields are bare for most of the year, aggregated annual PNV FAPAR is higher overall. Whilst this pattern may hold for rain-fed agriculture, in intensively irrigated areas the FAPAR of the managed vegetation can be much higher than of the PNV over the whole year, especially in arid and semi-arid areas (e.g., Nile Delta). This supplemental irrigation, plus the fact that total annual precipitation is the most important covariate, indicates that water availability/use efficiency is likely the main driver of FAPAR beyond natural conditions.

Maps of the s.d. of the prediction error (Fig. 14) as derived in the `ranger` package by using the `quantreg` setting (*Meinshausen, 2006*) provide useful
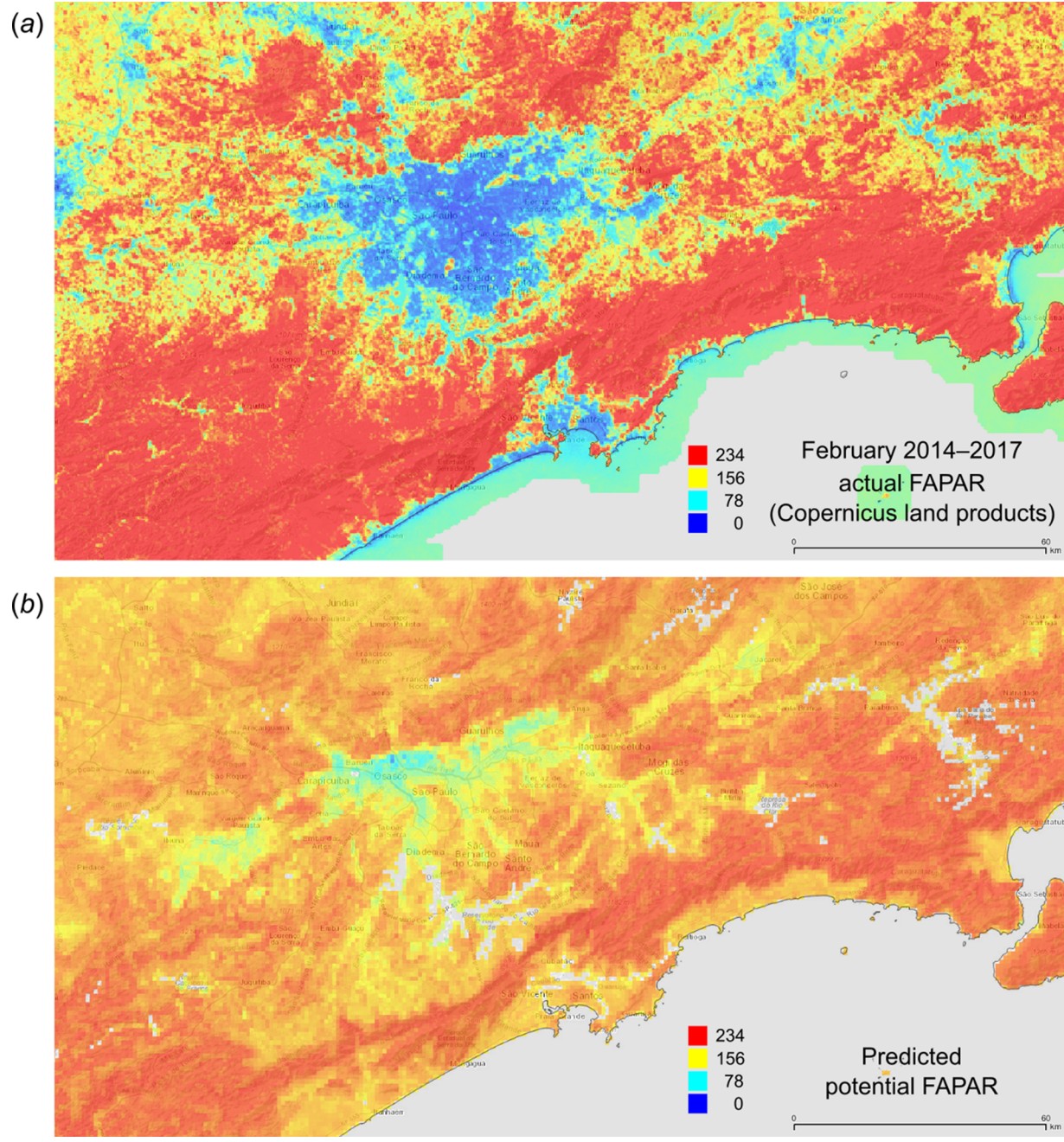

**Figure 12 FAPAR values for February based on the PNV samples: (A) actual (250 m resolution) and (B) predicted (one km resolution).** A zoom in area around the city of São Paulo in Brazil.

information about model quality, that is, where collection of additional points would maximize model improvement and which additional covariates could be considered. For example, the highest prediction errors for FAPAR for the month of August occurred in the transition areas between tropical forest and savanna areas, and in various biome transition zones in Asia.

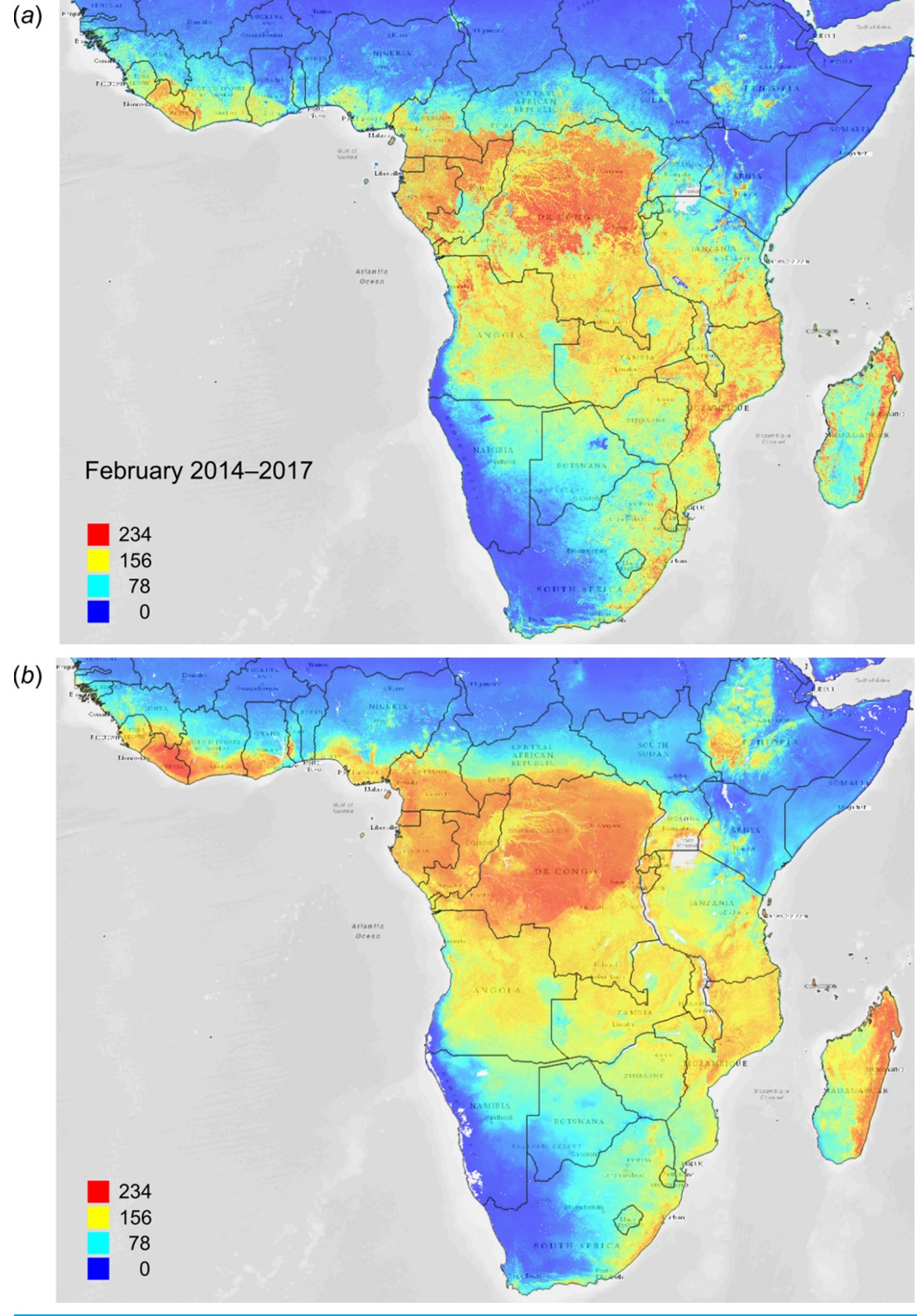

**Figure 13 FAPAR values for Subsaharan Africa: (A) actual (250 m resolution) and (B) predicted (one km resolution) potential FAPAR values for February.** Background map data: Google, DigitalGlobe.

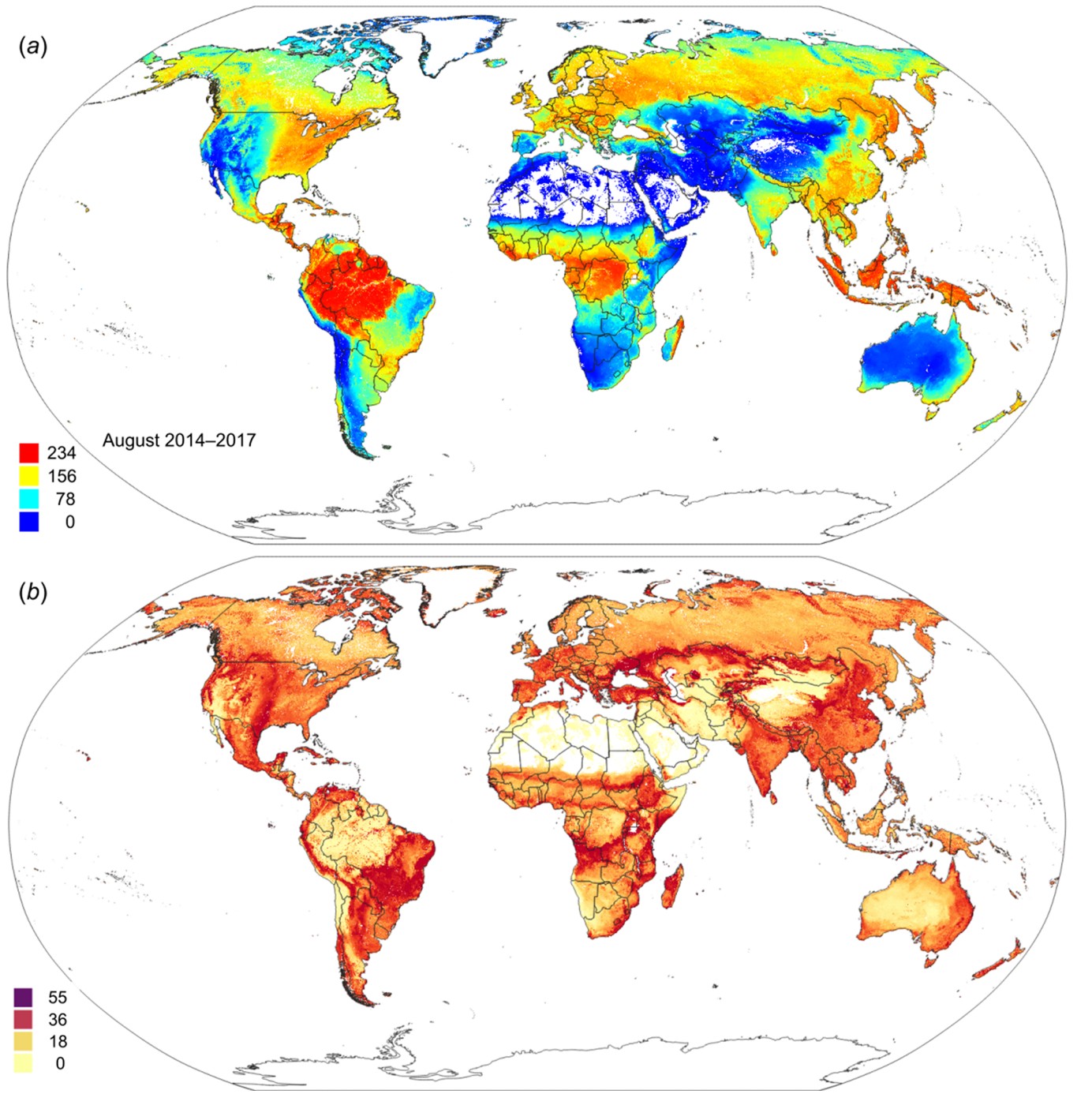

**Figure 14 Predicted global FAPAR values for August (A) and standard deviation of the prediction error for the map above (B).** To convert to percent, divide by 253.

## DISCUSSION

### Accuracy and reliability of produced PNV maps

Our results of modeling potential spatial distribution of global biomes, potential FAPAR and European forest tree taxa, show that relatively accurate maps of PNV

can be produced using existing data and publicly available environmental grids. In the case of the biomes and forest tree taxa case studies, RF consistently outperforms neural networks, gradient boosting, and similar MLA's. This is consistent with some other vegetation mapping studies (*Li et al., 2016*). However, RF and Cubist models perform equally well in the case of FAPAR. Accuracy assessment results of our work indicate improvement in product accuracy in terms of greater spatial detail and smaller classification error than found in the mapping products of *Levavasseur et al. (2012)* and *Tian et al. (2016)*.

Precipitation, temperature maps, and bioclimatic images are consistently the most important covariates in all three case studies. Currently available lithology/parent material maps are not indicated as significantly important covariates in any of the case studies. This may be because the existing lithologic map (*Hartmann & Moosdorf, 2012*) is not detailed enough, and/or because the differences in lithology/parent material are more important at finer resolutions/scales than those mapped here. Landform and lithology/parent material covariates may be important at local scales but, globally, vegetation distribution seems to be dominated by climate. This is not surprising since nutrient availability is also partially controlled by climate and partially by the vegetation itself. Upon visualization of the mapping products, however, it was noticed that the influence of topography is visible, especially in the maps of European forest tree taxa, suggesting that DEM derivatives are still important for mapping PNV.

We have also not considered any soil layers as inputs to modeling as these are also often predicted from similar climatic and remote sensing layers already used in our case studies as covariates. Moreover, most of the predictive soil mapping projects use RS images reflecting human induced changes, which we have tried to avoid as these are more relevant for mapping actual vegetation. For mapping of the potential managed vegetation, however, it would be probably more important to include also soil property/soil type maps into the modeling framework.

Further improvements in prediction accuracy of global biome may be limited due to:

1. BIOME reconstructions representing the vegetation of an area around a given site rather than at the exact point location, since the source of the pollen is on the order of 10–30 km around the site.
2. The ambiguity of reconstructions for about 10% of the sites, so that maximum accuracy of any prediction technique may not exceed 90% without additional observation data.
3. The fact that the BIOME reconstruction accuracy is known to be lower at ecotonal boundaries and in mountainous areas because of pollen transport issues, particularly the long-distance transport of tree pollen.
4. The BIOME data set is compiled from many regional reconstructions and all harmonization was done a posteriori, which may have introduced additional noise into the data.

So far, we have not explored opportunities for combining multiple MLA models based on validation data; that is, for doing ensemble predictions, model averages or model stacks.

Stacking models can improve upon individual best techniques, achieving improvements of up to >30%, with the additional costs including higher computation loads (*Michailidis, 2017*). In our case, the extensive computational load from derivation of models and product predictions had already obtained improved accuracies, making increasing computing loads further a matter of diminishing returns.

Our list of MLA models could also be extended. For example, we did not consider the use of support vector machines (*Li et al., 2016*), or the extreme learning machine algorithm (*Deo & Sahin, 2015*). Both have proven to be suitable for mapping vegetation distribution and quantitative properties of vegetation. Not all MLA methods are, however, suitable for large regression matrices, as the computing time can be excessive and hence parallelization options are crucial.

Our models of PNV FAPAR are based on simulated point data and the accuracy of how well models represent natural vegetation areas is dependent on the representativeness of the Protected Planet (http://protectedplanet.net) and Intact Forest Landscapes (http://intactforests.org) data. Also, many of the world's biomes such as the Mediterranean region and similar, have sustained high levels of human impact in the past and are perhaps under-represented in the Protected Planet (http://protectedplanet.net) data set. Nevertheless, our cross-validation results (Leave-Location-Out method) indicate a good match between training and validation points.

It would be useful to further explore what the performance of the models we used would be if we removed whole continents in the cross-validation process, or at least larger countries such as USA, China, Brazil, Australia, India, and/or the South African Republic. For biomes, spatial cross-validation showed a significant drop in accuracy; removing some larger countries from model training will likely also make difference. We did not explore effects of spatial proximity on mapping forest species and FAPAR as these are very dense point data sets. In addition, FAPAR training points were generated using simple random sampling, so spatial clustering should be non-existent.

*Fourcade, Besnard & Secondi (2018)* recently demonstrated that randomly chosen classical paintings can also be added to predictive modeling, and sometimes such models might be even better evaluated than models computed using real environmental variables. MLAs have even higher tendency to over-fit data and often perform very poor in extrapolation areas. These two remain the biggest drawbacks of using MLAs for species distribution modeling. It appears that the key to avoiding over-fitting or using non-realistic mapping accuracy measures, based on *Fourcade, Besnard & Secondi (2018)*, is in putting more effort in cross-validation (i.e., making it more robust and more reliable) and in ensuring that most important predictors and partial correlations can also be explained.

## Possible uses of the produced PNV maps

*Newbold et al. (2016)* argued that many terrestrial biomes today have transgressed safe limits for biodiversity, with grasslands being most affected, and tundra and boreal forests least affected. *"Slowing or reversing the global loss of local biodiversity will require preserving the remaining areas of natural (primary) vegetation and, so far as possible, restoring human-used lands to natural."* (*Newbold et al., 2016*) Roughly half of the difference of

around 466 billion tonnes of carbon can be attributed to the clearing of forests and woodlands, mostly for agricultural purposes (*Erb et al., 2017*). The other half of biomass carbon stock losses is derived from the management effects within a land cover class (*Erb et al., 2017*). The expansion of agriculture will probably continue in the future, leading to decreased biodiversity and soil degradation (*Mauser et al., 2015*; *Molotoks et al., 2017*). On the other hand, *Griscom et al. (2017)* identify reforestation (e.g., biomass restoration) as the largest natural pathway to hold global warming below 2 °C. In that context, accurate maps of PNV could become increasingly useful for assessing the level of land degradation/biomass shortfall relative to the potential of a site. Such information can also inform selection of optimal steps toward restoring biomass stocks in managed vegetation in ways that better reflect the PNV FAPAR in those areas.

Other uses of PNV maps include assessing the land potential, that is, land use efficiency given the difference between actual and potential vegetation. Consider, for example, a location in southern Spain called *"Altiplano Estepario,"* which has been identified by the Commonland company (http://commonland.com) and partners as a landscape restoration site. Figure 15 shows results of a spatial query for this location and values of our PNV and PNV FAPAR predictions, in comparison to the actual land cover and actual FAPAR images. The figure shows that the actual FAPAR is as good as PNV FAPAR in February and March but that differences are large in the summer months. Overall, the median and upper FAPAR for this specific location are only 51% of the PNV FAPAR, so we can say that this site is currently operating at 51% of the predicted FAPAR capability under PNV. This comparison should also consider that our estimates of FAPAR come with an RMSE of ± 0.085. Furthermore, as landscape restoration efforts have recently begun on this site—this work suggests that it ought to be possible to: (a) identify priority areas of PNV FAPAR shortfall, (b) use this information to inform in part the type of restoration strategies used, and (c) monitor the progress of restoration efforts in monthly time steps over several decades. Such practical measurement, monitoring and verification efforts are required to mobilize further investment in this emerging sector.

Our PNV maps could also be used to estimate soil carbon sequestration and/or evapotranspiration potential, and gains in net primary productivity assuming return of natural vegetation (Fig. 16). Furthermore, by combining various estimates of potential natural and managed vegetation, one could design the optimal use of land both regionally and globally. *Herrick et al. (2013)*, for example, provide a theoretical framework for estimating land potential productivity which could theoretically connect all land owners in the world to share local and regional knowledge.

Maps of PNV for European tree species could also be used as a supplement to the distribution and ecology of tree species produced by *San-Miguel-Ayanz et al. (2016)* and *Brus et al. (2012)*. Species such as *Carpinus orientalis*, *Cupressus sempervirens*, *Prunus mahaleb*, *Sorbus domestica* are all predicted with TPR <0.5 indicating critically poor accuracy. Possible reasons for such low accuracy are problems with representation of training points and somewhat too broad ecological conditions, especially if a species follows some other more dominant tree species that have wide ecological niche. These maps should probably not be used for spatial planning.

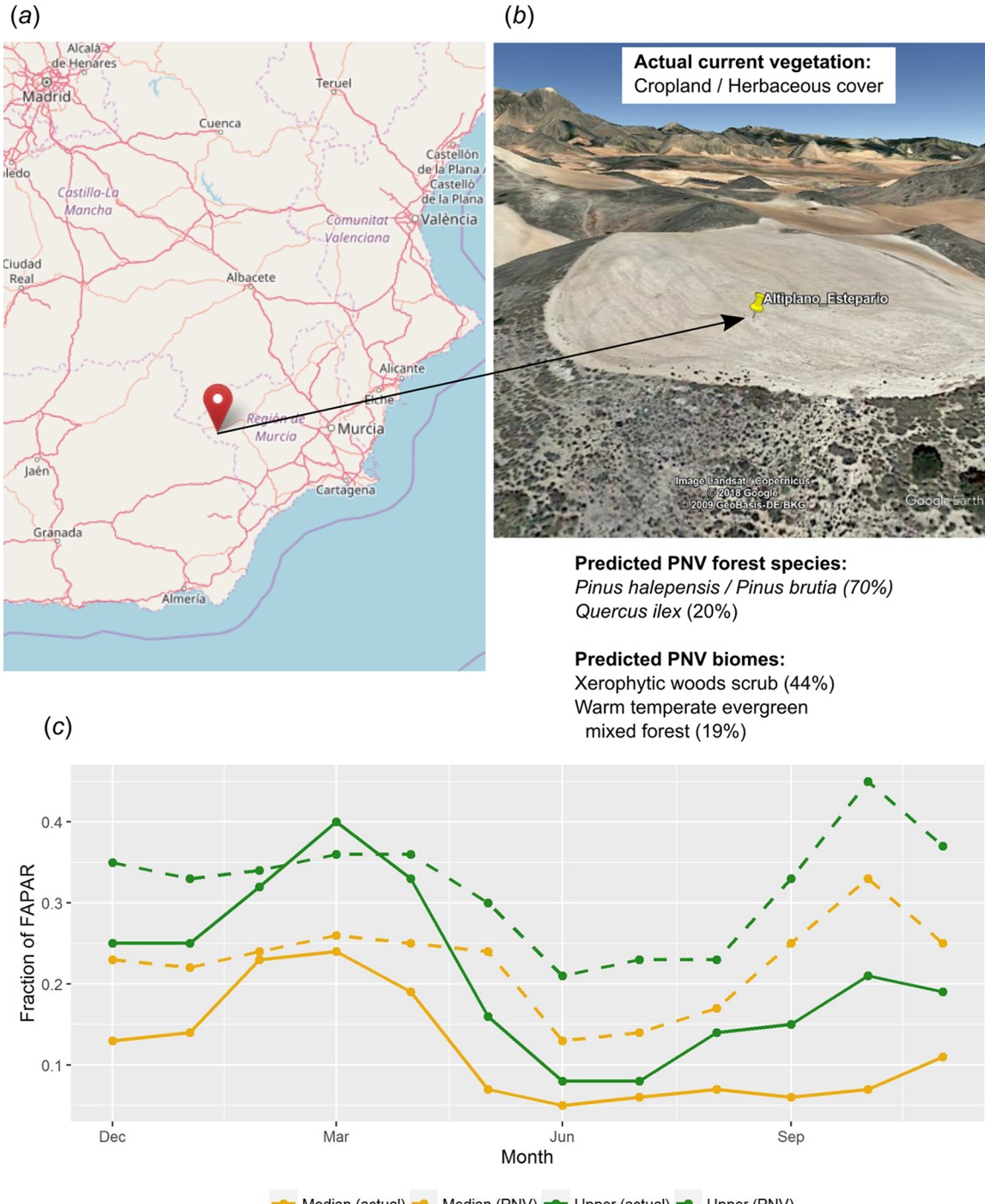

(a)

(b)

**Actual current vegetation:**
Cropland / Herbaceous cover

Altiplano_Esteparo

Image Landsat / Copernicus
© 2018 Google
© 2009 GeoBasis-DE/BKG
Google Earth

**Predicted PNV forest species:**
*Pinus halepensis / Pinus brutia* (70%)
*Quercus ilex* (20%)

**Predicted PNV biomes:**
Xerophytic woods scrub (44%)
Warm temperate evergreen
 mixed forest (19%)

(c)

Fraction of FAPAR

Month

● Median (actual) ● Median (PNV) ● Upper (actual) ● Upper (PNV)

**Figure 15 Example of comparison between the actual land cover and actual FAPAR curves and our predicted potential natural vegetation (PNV) and predicted PNV FAPAR curves.** According to our results, this location (A and B) in southern Spain (latitude = 37.938478, longitude = −2.176692) currently utilizes 51% of the predicted FAPAR capability under PNV, indicating a substantive short fall in on-site photosynthetically active biomass (C). Background map (A) source: OpenStreetMap; landscape view (B) map data: Google, DigitalGlobe.

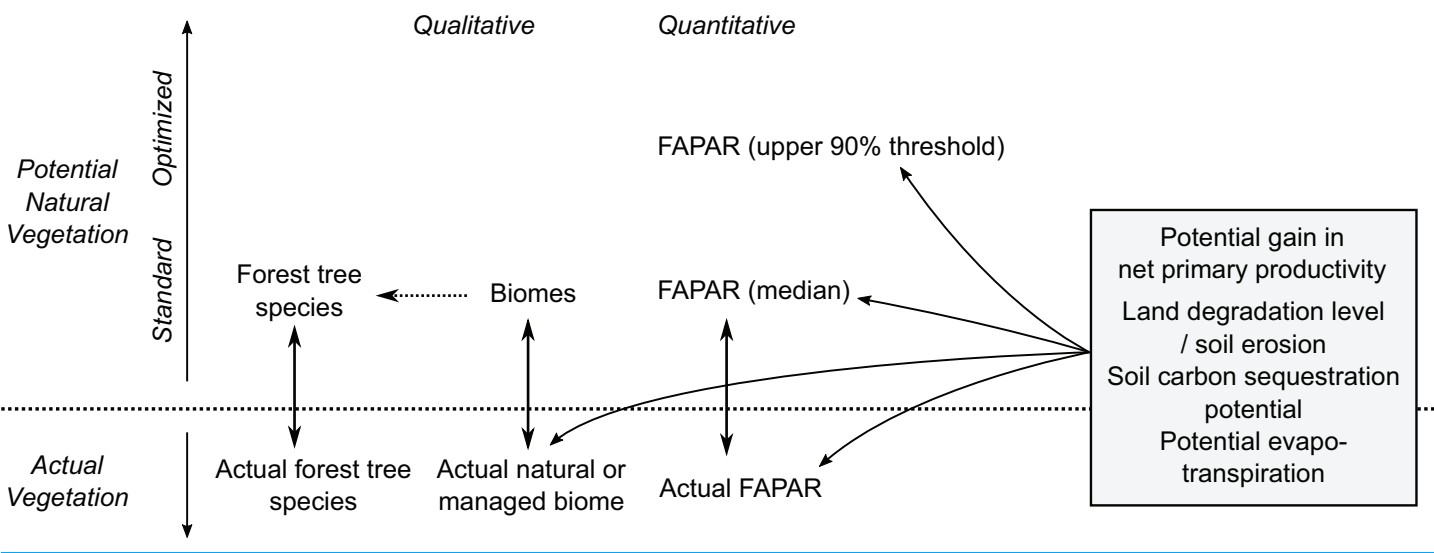

**Figure 16 Some possible uses of maps of potential natural vegetation.**

Potential natural vegetation for European tree species analysis could be made even more quantitative so that even predictions of dendrometric properties of tree species could be produced using similar frameworks. Also, similar PNV mapping algorithms could be used to map the potential canopy height based on the previously estimated map of the global canopy height (*Simard et al., 2011*).

## Technical limitations and further challenges

Running machine learning algorithms on larger and larger data is computationally demanding; however, by using fully parallelized implementation of RF in the ranger package, we were able to produce spatial predictions within days. Model fitting and prediction using EU Forest and GBIF data (1.5 million training points) was, however, very memory and time consuming and is not recommended for systems with <126 GiB RAM. In our case, model fitting took several hours even with full parallelization, and final models were >10 GiB in size. Prediction of probabilities took an additional 5–6 h with the current computational set-up. In the future, scalable cloud computing could be used to overcome some of these computational limits. Machine learning will in any case continue to play a central role in analyzing large remote sensing data stacks and extracting useful spatial patterns (*Lary et al., 2016*).

With enough computing capacity, one could theoretically use all 160 million records of distribution of plant species currently available via GBIF (*Meyer, Weigelt & Kreft, 2016*) and from other national inventories to map global distribution of each forest tree species. In Europe the list is very short; globally this list could be quite long (e.g., 60,000 species). The primary problems of using GBIF for PNV mapping will remain, however, as these are primarily due to high clustering of points and under-representation of often inaccessible areas with very high biodiversity (*Yesson et al., 2007*; *Meyer, Weigelt & Kreft, 2016*). GBIF records have been shown in the past to give biased results (*Escribano, Ariño & Galicia, 2016*), so that spatial

prediction methods that account for high spatial clustering, that is, bias in training point representation in both space and time; would need to be developed further to minimize such effects.

## CONCLUSIONS

Although PNV is a hypothetical concept, ground-truth observations can be used to cross-validate PNV models and produce an objective estimate of accuracy. As the prediction accuracy becomes more significant, the reliability of the PNV maps increases. Our analyses show that the highest accuracy for predicting 20 biome classes is about 68% (33% with spatial cross-validation) with the most important predictors being total annual precipitation, monthly temperatures, and bioclimatic layers. Predictions of 73 forest tree species had a mapping accuracy of 25% and with average TPR of 0.69, with the most important predictors being mean annual and monthly temperatures, elevation, and monthly cloud fraction. Regression models for FAPAR (monthly images) were most accurate with R-square of 90% (Leave-Location-Out CV) and with the most important predictors being total annual precipitation, MODIS cloud fraction images, CHELSA bioclimatic layers and month of the year, respectively. Machine learning can be successfully used to model vegetation distribution, and is especially applicable when the training data sets consist of a large number of observations and a large number of covariates. Extending the coverage of observations of natural and managed vegetation, including through making new ground observations, will allow regular improvements of such PNV maps.

## ACKNOWLEDGEMENTS

This research is a contribution to the AXA Chair Programme in Biosphere and Climate Impacts and the Imperial College initiative on Grand Challenges in Ecosystems and the Environment (ICP). Authors are grateful to *Karger et al. (2017)* for maintaining the CHELSA Climate images, US agencies NASA and USGS for distributing high resolution images of Earth's atmosphere and the European Copernicus Land program. We are grateful to *Mauri, Strona & San-Miguel-Ayanz (2017)* for sharing the EU-Forest—a high-resolution tree occurrence dataset for Europe. We are also grateful to the Open Source software developers of the packages `ranger`, `xgboost`, `caret`, `raster`, `GDAL`, `SAGA GIS` and similar, and without which this work would have not be possible.

### Funding

Funding was provided by The Nature Conservancy and the Doris Duke Charitable Foundation. Sandy P. Harrison was supported by the ERC-funded project GC2.0 (Global Change 2.0: Unlocking the past for a clearer future, grant number 694481) and from JPI-Belmont Forum via NERC for the project "PAleao-Constraints on Monsoon Evolution and Dynamics (PACMEDY)." The funders had no role in study design, data collection and analysis, decision to publish, or preparation of the manuscript.

## Grant Disclosures

The following grant information was disclosed by the authors:

Nature Conservancy and the Doris Duke Charitable Foundation.

ERC-funded project GC2.0 (Global Change 2.0: Unlocking the past for a clearer future, grant number 694481).

JPI-Belmont Forum via NERC for the project "PAleao-Constraints on Monsoon Evolution and Dynamics (PACMEDY)."

## Competing Interests

Tomislav Hengl and Ichsani Wheeler are employed by Envirometrix Ltd.

## Author Contributions

- Tomislav Hengl conceived and designed the experiments, performed the experiments, analyzed the data, contributed reagents/materials/analysis tools, prepared figures and/or tables, approved the final draft.
- Markus G. Walsh conceived and designed the experiments, performed the experiments, analyzed the data, contributed reagents/materials/analysis tools, authored or reviewed drafts of the paper, approved the final draft.
- Jonathan Sanderman conceived and designed the experiments, performed the experiments, analyzed the data, authored or reviewed drafts of the paper, approved the final draft.
- Ichsani Wheeler conceived and designed the experiments, analyzed the data, contributed reagents/materials/analysis tools, prepared figures and/or tables, approved the final draft.
- Sandy P. Harrison analyzed the data, prepared figures and/or tables, authored or reviewed drafts of the paper, approved the final draft, cleaned up biome data set.
- Iain C. Prentice analyzed the data, prepared figures and/or tables, authored or reviewed drafts of the paper, approved the final draft, cleaned up biome data set.

## Data Availability

Harvard Dataverse, Global Maps of Potential Natural Vegetation at one km resolution: http://dx.doi.org/10.7910/DVN/QQHCIK

All code used to generate the maps is available at: https://github.com/Envirometrix/PNVmaps.

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
