# Peer review of "Global mapping of potential natural vegetation: an assessment of machine learning algorithms for estimating land potential"

_PeerJ, doi:10.7717/peerj.5457_

## Round 0.1 · original submission · Minor Revisions

· Academic Editor

Minor Revisions

The four referees suggest that the submission may be publishable, but only after many minor revisions have been made to your manuscript. Therefore, I invite you to respond to their comments and revise your manuscript.

Reviewer 1 ·

Basic reporting

no comment

Experimental design

no comment

Validity of the findings

no comment

Additional comments

This paper is well structured and written. It compares four machine learning methods for potential natural vegetation (PNV) mapping from climate and environmental covariates. The results show that the random forest performs the best, which is coherent to many other findings solving classification/regression issues. I have a minor comment on this paper.

Many of MODIS FPAR is from the vegetation that has been affected by human activity. In the training stage, the author applied the WDPA and IFL mask to select samples with minimum affection by human activity. These two masks make the training pool data spatially unevenly distributed. Please clarify whether random sampling or stratified sampling is applied and discuss its possible effect on the results.

·

Basic reporting

This manuscript is aimed to produce PNV maps that are both more detailed, richer in information, based on multiple machine learning algorithms.
The writing is clear enough, however the manuscript contains a number of spelling and grammatical errors which should be carefully revised before acceptance.
The literature is suitable for the purpose of the article.

Experimental design

no comments

Validity of the findings

The data are robust and results compatible with conclusions.
It is necessary to present evaluation of TPR outliers in the Table 2. (e.g., Carpinus orientalis and Cupressus sempervirens).
In the revision round, it would be interesting to compare the performances of multiple MLA.

Additional comments

1. As the title of this manuscript listed “Global Mapping of Potential Natural Vegetation: An Assessment of Machine Learning Algorithms for Estimating Land Potential”. Content of PNV and FAPAR is too much in the manuscript. Authors should focus on the effect on wind erosion potential by land use/cover changes and land management practices.
2. The text is extensive and repetitive in many parts. Thus, I recommend rewrite abstract and introduction. The conclusions can be improved.
3. There are some sentences which are difficult to understand, for instance:
Page 4/lines 105-106: For efforts aimed at land restoration ....
Page 9/lines 239-240: We use about 3× more training points ....
Page 12/lines 344-346: TPR values range from 0 to 1 where 1 ....
Page 16/lines 394- : Results of the spatial Cross-Validation, as implemented ....
4. The figures should be renumbered accordingly such that they are introduced in the correct sequence (e.g., Fig 1). You did not mention Fig 1a in the manuscript.
5. The time stamps in Figure 9 are hard to read, as are the lines in Figure 10, which makes them difficult to interpret. There seems to be no explanation in the Figure legend or any values which describe the distribution of the tree species, it may be helpful to put these directly in the Figure.
6. The reference citation in the title of Table 2 is inappropriate. I recommend deleting the citation and placing the correlated content information in the text. The Species name of forest tree taxa should be placed in italic type.
7. The spatial resolution of Input data (environmental covariates) should be illustrated in detail.
8. Line 287 The resampling methods of nearest neighbor interpolation, bilinear interpolation, and cubic convolution interpolation were commonly used. In this study, which one were used?
9. The model inputs seem to be complicated. It would be better to have a schematic picture that shows the model framework.
10. The abbreviation of Scaled Shannon Entropy Index has two forms (EIs, SSEI, NDVI and some other technical terms) should be uniformed, and the corresponding full name should be given. Work on text clarification and formatting, improve figures, list of the abbreviations would be useful in this case to easily understand the text.
11. The quantitative comparison of results of modeling potential distribution of tree species in Europe should be illustrated, not only the visual comparison.
12. The calculating equations of RMSE and ME should be listed as standard equations with number (Page 13/line 366). The variables in the equations of them are not explained.
13. More discussions on the ecological gradients should be added to the manuscript.
14. The detection and assessment of PNV is critical for the prevention of environmental deterioration especially in arid and semi-arid areas. Hence, I propose to consider the special region.
15. The altitude, topography and soil types might have a certain impact on the PNV. Due to the lack of information it is not clear the need of pooling all data together. I am wondering if by comparing different climate (soil/topographical) regions and analyze the response of the methodology under these scenarios would not produce better results. In any case, this discussion was missing and need to be clarified.
16. The challenges and limitation in mapping PNV using MLA should be further highlighted in the discussion.
17. Line 416 “Correlation analysis (predicted distribution maps) indicates that…” is not clear enough. Which fig (did you mean fig. 6)? There should be more details here.
18. In general, the machine learning algorithm with more parameters or hyper-parameters could lead to more complicated training, though it has better accuracy. A good algorithm should have good accuracy and meanwhile, include less trained parameters. It looks that RF has more complicated hyper-parameters. More discussions on this point should be added to the paper.

Reviewer 3 ·

Basic reporting

This paper is well written and fits into the scope of this journal. This study contributes to the current knowledge of natural vegetation mapping using data-driven approaches. The methodology presented in this study would help other researchers to replicate the methods and apply to other parts of the world, provided all the data used in this study are available.

Experimental design

Modeling of natural vegetation across the globe using statistical and machine learning models is very useful for biodiversity management, and help quantify the effect of climate change and human activities on existing vegetation. Empirical/data-driven models can serve as surrogates to computationally intensive physical/numerical vegetation models. The authors did a commendable job in collecting the huge amount of data and developing models that bridge the gap between existing models and present knowledge of vegetation mapping.

Validity of the findings

Conclusions are well presented and answer the research questions.

Additional comments

1. Line 23: PNV is already spelled out in line 20, so just say PNV here.
2. Could you please elaborate the 'time' label on x-axis?
3. Fig. 12: Bottom figure: It should be predicted FAPAR.
4. Please include a section on assumptions that are considered in this study or as a section describing limitations of the modeling framework. I would keep all the assumptions in one place to get a complete picture of what data are available and what are the limitations.
5. Fig. 14: Please elaborate 'To convert to percent divide by 253'.
6. Did you run your models on GPU, if yes give the specifications.
7. How long does it take to run each model (random forest, neural network etc.)? Please provide a comparison of computation time, performance, parameter sensitivity etc. maybe in the form of table for better understanding.
8. Could you please provide a comparison of the best-predicting model results to ground-truth observations?
9. What are the percentage of relevance of the most important predictors like precipitation, temperatures etc.? Can you calculate those from connection weights of the neural network?
10. It would be nice to refer some of the recent machine learning papers in related field.
For example:
Deo, R.C. and Şahin, M., 2015. Application of the extreme learning machine algorithm for the prediction of monthly Effective Drought Index in eastern Australia. Atmospheric Research, 153, pp.512-525.
Sahoo, S., T. A. Russo, J. Elliott, and I. Foster (2017), Machine learning algorithms for modeling groundwater level changes in agricultural regions of the U.S.,Water Resour. Res.,53, 3878–3895, doi:10.1002/2016WR019933.
Okujeni, A., van der Linden, S. and Hostert, P., 2015. Extending the vegetation–impervious–soil model using simulated EnMAP data and machine learning. Remote Sensing of Environment, 158, pp.69-80.
Li, X., Chen, W., Cheng, X. and Wang, L., 2016. A comparison of machine learning algorithms for mapping of complex surface-mined and agricultural landscapes using ZiYuan-3 stereo satellite imagery. Remote sensing, 8(6), p.514.

·

Basic reporting

The work presents a comprehensive study of the effectiveness of different machine learning approaches (in particular, Artificial Neural Networks, Random Forest, Generalized Boosted Regression and K-NN) applied to the global mapping of Potential Natural Vegetation (PNV). In general terms, it is an interesting contribution. It is well structured, follows a clear methodology and provides a wide experimentation. The writing is good, which is accompanied by visual content that facilitates its understanding. The paper meets all the requirements to be accepted. However, there are minor modifications that may greatly improve the dissemination of the performed research

Experimental design

Although the experimentation is appropriate, the paper would gain in interest if it would provide a more in-depth discussion about the motivation that led to select the implemented algorithms, as well as their hyphotetized benefits/drawbacks when applied to PNV mapping.

Validity of the findings

The extensive and methodical experimentation proves the effectiveness of the selected methods. However, the paper would gain in clarity if the achieved results were correlated with the characteristics of the algorithms (from the ML and pattern recognition point of view).
On the other hand, I suggest to enhance the criteria considered for estimating the best TPR, since the ROC curve covers the spectrum of all possible values based on calibration, and a simple average may mislead the results interpretation. Maybe the best Youden index could fit with the experimental purposes.

Additional comments

The paper provides an interesting contribution, which a well-described research methodology and an extensive experimentation. Hence it meets all the requirements to be accepted

---

## Round 0.2 · accepted · Accept

· Academic Editor

Accept

I can confirm that your revision is satisfactory, and I am pleased to inform you that your paper has been accepted for publication.

#